# Large-Scale Land Acquisition and Household Farm Investment in Northern Ghana

Abdul-Hanan Abdallah [1,*], Michael Ayamga [2] and Joseph Agebase Awuni [3]

1 Department of Agricultural and Food Economics, University for Development Studies, Tamale P.O. Box TL1350, Ghana
2 School of Applied Economics and Management Sciences, Department of Applied Economics, University for Development Studies, Tamale P.O. Box TL1350, Ghana
3 School of Applied Economics and Management Sciences, Department of Economics, University for Development Studies, Tamale P.O. Box TL1350, Ghana
* Correspondence: aahanan@uds.edu.gh

**Abstract:** Many studies have investigated the effects of large-scale land acquisition (LSLA) on livelihood, while the effects of LSLA by different actors on investment decisions and levels of investment have largely gone without academic scrutiny. Consequently, information concerning the implications of LSLA by actors on investment is scarce in the literature pertaining to policy. Drawing on information from 664 households selected through a multistage sampling technique, this study examined the relationship between direct and indirect exposure to LSLA by domestic and foreign entities and investment in land-improving techniques. The results show a bi-directional relationship between LSLA and household farm investments. While direct and indirect exposure to LSLA by domestic and foreign entities dissipates some forms of farm investments, the reverse causality is also possible where some household farm investments discourage direct and indirect exposure to LSLA by domestic and foreign entities. The results also revealed that LSLA by domestic and foreign entities dissipates investment in all levels of land- and yield-improving techniques, and even in the presence of a high perception of tenure security. Thus, the provision of legal ownership of land to farmers can provide insurance for investments in all land-improving techniques. Government can also step up the fertilizer subsidy program to enable households to increase investment to avoid further exposure to LSLA.

**Keywords:** land acquisition; farm investment; land- and yield-improving techniques; Ghana

## 1. Introduction

Smallholder adoption and investment decisions on land quality and yield improvement techniques have been long-term policy objectives of most governments and policy makers in sub-Saharan Africa [1], and in Ghana in particular [2]. This is because land quality and yield improvement techniques play important roles in sustainable development [3,4]. Aside from conserving resources and being environmentally friendly, land quality and yield improvement techniques are also economically profitable [1]. Thus, improving adoption investment decisions using such techniques will not only maintain soil fertility, but improve yields, productivity, food security, and sustainable development. However, at the heart of smallholder adoption and investment decisions is secured land access. Aside from being directly linked to investment in capital and land improvement techniques (see [5]), secured land access promotes access to incentives [6] for investment in land improvement techniques. Thus, adoption and investment decisions in land quality and yield improvement techniques are affected by any issue faced by smallholder land access.

Large-scale land acquisitions (LSLA) by corporations for plantation agriculture is a topical issue concerning farmland access. Although such acquisitions have existed since

the colonial period, the practice skyrocketed following the 2007 to 2008 multiple crises (i.e., food, energy/fuel, climate, and financial), and thus places smallholder adoption and investment decisions at center stage for researchers and policy makers. The central issue of such controversy is the effect of LSLA on livelihoods, including smallholder adoption and their investment decisions. Some (e.g., [7]) have argued that LSLA can bring about job creation, technology spillover, and smallholder investment in land- and yield-improving techniques, while others (e.g., [8]) argued that technology spillovers and investment in land- and yield-improving techniques is often low, since such acquisitions often fail to generate the desired employment opportunities among smallholders.

In an attempt to resolve the controversies surrounding LSLA, several empirical studies have investigated its implications on livelihoods (e.g., [9–13]). However, questions regarding the effect of LSLA by different actors on investment in land- and yield-improving techniques have been rarely answered. Thus, information on investment and level of investment in land- and yield-improving techniques in the wake of LSLA by different actors is missing in the literature, even though such information could be highly relevant for policymakers in Africa. This paper addresses the implications of LSLA by different actors on investment in land- and yield-improving techniques in Ghana. Currently, the only studies that were close to this kind of analysis include [14–17]. However, these studies did not consider farm investment implications of LSLA by the actors involved.

Moreover, proper analysis of the implication of LSLA on land improvement techniques requires accounting for the potential endogeneity of LSLA. However, with the exception of [15], none of these studies focused on the potential endogeneity of LSLA in examining its implication on households' investment in land and yield improvement techniques. Furthermore, the results of these studies (i.e., [14–17]) are mixed, and we do not know the exact implication of LSLA on households' investment in land- and yield-improving techniques. For instance, while [14,15] and [17] found evidence of increased investment in land improving practices, [16] found decreased investment in the same. This paper fills these gaps by analyzing the impact of LSLA by different actors on investment in land- and yield-improving techniques in Ghana. The novelty of this study exists in three aspects.

First, the study examines the implications of LSLA on household farm investment with special reference to different actors involved in LSLA, and has a particular interest in the exact impact they have on land- and yield-improving techniques. Available information on LSLA (e.g., [18,19]) showed that different actors including chiefs, home, and foreign country governments, financial entities, large-scale agro-processing industries, traditional agricultural or agro-industrial operators, public and private investors, or domestic and foreign players are now involved in LSLA, even though the size of land controlled by these actors are dissimilar. Thus, adoption and investment decisions and their relationship with LSLA by different actors will be of much more interest to stakeholders in both land management and agricultural development in Africa. We specifically focused on how farm investments are affected by direct and indirect exposure to an LSLA by domestic and foreign entities (detailed in the next section). To the best of our knowledge, this kind of study has not been conducted in Ghana or elsewhere.

Second, the study extends the analysis to the implication of LSLA on level of investment in land and yield improvement techniques, as there are several investment levels that LSLA affects in different ways. Farm investment is defined as an inter-temporal phenomenon in which households expect to reap a stream of future returns/profit—in the form of increased value of land, productivity, and assets—by spending/supplying resources on/to land [20]. Generally, household farm investment decisions focus on three main activities, including capital equipment, land improving techniques, and nonagricultural activities and assets [5]. However, in this study, our focus is on how direct and indirect exposure to an LSLA by domestic and foreign entities affect land and yield improvement techniques, including wells (W), local catchments (LC), drip irrigation facilities (DIF), inter-cropping with nitrogen fixing crops (I), minimum tillage (MT), residue retention (RR), NKP,

Sulphate of Ammonia (SOA), and Urea (U). We focused on these techniques as they are directly affected if the household is exposed to LSLA.

Third, endogeneity of LSLA could be a nontrivial issue in estimating the effect of LSLA on investments in potentially interrelated land and yield improvement techniques. This study extends the control function approach proposed by [21] and applies extended regression models [22] to detailed household level survey data from the 2017–2018 cropping season, in order to account for potential endogeneity arising from both observed and unobserved heterogeneity and assess the effects of LSLA on investment in interrelated techniques, as well as the level of investment in these techniques.

The rest of the paper is structured as follows. In Section 2, a historical overview of large-scale land deals in Ghana is presented. Section 3 presents a review of the agricultural policies that triggered interest in agricultural investments. In Section 4, the methodology is presented. Section 5 presents the results and discussions, while conclusions are presented in Section 6.

## 2. Definition and Historical Overview of Large-Scale Land Acquisition in Ghana

Several definitions for large-scale land acquisition (LSLA)—a phenomenon commonly referred to as land grabbing—abound in the literature (e.g., [18,23]). However, as the study is situated in Ghana, a definition by the Lands Commission of Ghana—one of the lead agencies in charge of land management in Ghana—is adopted as the operational definition for LSLA.

According to the Lands Commission, an LSLA involves the acquisition of land that covers a land area of around 20.23 hectares or more [24]. Such acquisitions are usually characterized by investments in the production of mangos, rice, groundnuts, soya beans, millet, jatropha carcass, and onion for sale in either domestic or international markets. Furthermore, such acquisitions refer to the guidelines of the Land Commission of Ghana, or the code of conduct proposed by [23], for transparency, respect of human rights, sustainability of benefits, and the environment in acquiring or leasing land on a large scale. An LSLA by domestic entities includes all forms of acquisitions that are wholly perpetuated by domestic entities [25]. On the other hand, an LSLA by foreign entities is defined to include all forms of acquisition that are perpetuated by foreign entities (ibidem). Direct exposure to LSLA by domestic or foreign entities involves losing farmland, labor, or forest resources due to enclosures by domestic or foreign entities, while indirect exposure to LSLA by domestic or foreign entities involves living in an affected community, losing access to uncultivated land, or having limited land due to enclosures by domestic and foreign entities (ibidem).

In Ghana, LSLA and its characteristics are far from new or unique; there have been instances of such acquisitions, and attempts to make such acquisitions, throughout history. For instance, aside from the seizing of vast areas of land throughout the territorial wars of the pre-colonial period [26], there were several instances of attempts to acquire vast land during the colonial period. Such attempts to acquire land on a large scale are deeply rooted in the Crown Lands Bill of 1894.

The Crown Lands Bill of 1894 was designed to repose the power of traditional authorities over lands described as 'wasted lands' to the colonial government, to regulate concessions of mining and timber rights over vast tracts of land by these authorities to European mining and timber companies. However, attempts to pass the bill received resistance from intellectuals who argued that the people of Ghana were capable of managing their land without supervision, direction, or legislation from the colonial government. The role of newspaper editors in the resistance was significant, as they argued that the bill was claiming the 'wasted lands' for the use of the colonial government. Traditional authorities, on the other hand, petitioned the colonial government with historical documents (e.g., Bond of 1844 and the 1874 Proclamations) that never claimed any land outside of the forts and castles of the colonial administration. Given the strength and convincing nature of the resistance, the colonial administration conceded that the bill could not be passed without amendments. Thus, in 1897, the bill was modified. The modifications included the

abandonment of the intent to vest 'wasted lands' into the colonial government. Instead, the bill indicated that 'public land' was to be administered by the colonial government for the good of everyone. This implied that the right of ownership of land was no longer automatically recognized until a grant of land certificate was issued by the colonial government. However, attempts to pass such a bill were again resisted, with the argument that it still placed too much power in the hands of the colonial government [27]. This led to the failure of the second bill of 1897. Thus, the current wave of LSLA is not new, but a repetition of similar historical strategies.

The difference, however, is that the historical strategies abated current authorities in transferring land on a large scale to investors. In other words, the current wave of acquisition is driven by legislative interference that failed in the past. Both traditional and state authorities have labeled some lands as 'unused' or 'marginal', but have succeeded in transferring such lands to investors. For example, in the Agogo Traditional Council of Southern Ghana, land considered to be "bush" or "marginal" by chiefs was transferred to ScanFarm Ghana Ltd. [28]. In the northern region of Ghana, a report regarding unused land led to its acquisition by Biofuel Africa Ltd. [28]. In addition, state laws and policies now facilitate LSLA, even though no known law and policy succeeded in facilitating LSLA in the past. In particular, agricultural policies play a central role in the recent wave of LSLA, and are discussed in detail in the next section.

Additionally, the current wave of acquisition differs from past attempts in terms of the trends and actors involved. Available statistics from the Land Matrix database regarding LSLA in Ghana (Table 1) show that out of a total of 1,502,483 ha intended to be acquired in Ghana, 551,091 ha has been acquired so far, and of the total acquired, about 95,873 ha is under domestic actors, while 455,218 ha is under transnational actors [19].

**Table 1.** Large-Scale Land Acquisition in Ghana.

| Deal Scope | Land Intended to Be Acquired (ha) | Land Acquired |
| --- | --- | --- |
| LSLA by domestic actors | 186,163.00 | 95,872.79 |
| LSLA by transnational actors | 1,316,320.40 | 455,218.40 |
| Total | 1,502,483.00 | 551,091.18 |

Source: [19].

Given that agriculture, which is the main source of livelihood for most households in Ghana, largely depends on land [29], the livelihoods of agricultural households are likely to be affected by LSLA by foreign and domestic entities. However, studies investigating the implications of LSLA by these actors on livelihoods are scarce in the literature. In particular, empirical studies regarding the effect of LSLA by domestic and foreign actors on livelihood strategies, including households' investment in land improvement techniques, are limited, and such information could be very useful for land use and development policies. This study focuses on farm investment effects of LSLA in northern Ghana, with specific reference to LSLA by domestic and foreign actors.

## 3. Agricultural Policies and the Rise of LSLA in Ghana

Since 2003, the global economy has experienced multiple crises, including food, oil, and financial crises [25]. This has triggered several challenges in the food systems of most economies [7]. Food insecurity, in particular, became a major challenge for most economies [29], and thus, prompted some food import-dependent countries to reconsider their policies for food security and sustainable development [7].

On the other hand, agriculture is a strong option for improving food security, as well as the 2030 agenda for sustainable development [30]. As a result, the sector has become the focus of most policy efforts to guide local and international investments to enhance food security. However, much as such policy efforts have enhanced food security, they have also promoted LSLA.

At the global level, policy efforts including the principles for responsible investment in agriculture (RAI) [31] and the code of conduct (CoC) for large-scale land acquisitions and leases [23] were developed to guide investment in agriculture to improve food security. Yet the propositions of the RAI and CoC promote LSLA by softening the perceptions of opponents about the dangers of LSLA on peasant livelihoods, and thus make it possible for governments to intervene less intrusively and more efficiently in society (see [32] for detailed discussion).

Another exemplary effort was the Rio+20 Summit by the UN on sustainable development, where commitments were made to enhance food security [33]. These efforts were also evidenced in the Maputo Declaration and New Economic Partnership for African Development and its Comprehensive Africa Agricultural Development Programme, in which African governments pledged to increase productive land under sustainable land management, and to also set aside at least 10% of the national budget for agriculture [1]. However, as part of the mechanisms that enhance food security, the Rio+20 Summit pledged to adopt voluntary guidelines for land investment to ensure transparency in bulk land acquisitions [33].

In Ghana, the first and second policies for Food and Agriculture Sector Development (FASDEP I and II) [2] and the Medium-Term Agricultural Sector Investment Plan [34] were both developed and implemented in accord with the Maputo and Malabo Declarations to increase productivity and food security.

Additionally, the Block Farm Programme (BFP) [35] and the Ghana Commercial Agriculture Project (GCAP) [36] were developed to promote the commercialization of agriculture to enhance food security. In addition to these efforts, the Land Administration Project was jointly implemented by the World Bank and Ghana's government to stimulate economic development and reduce poverty [37]. Some traditional authorities also transferred land on a large scale to investors to create development opportunities for local occupants [28].

However, these efforts have promoted LSLA in such a way that different investors, including domestic and transnational agribusiness investors, are eagerly acquiring large tracks of land to help secure food and energy needs for the future [32]. The GCAP, for example, encouraged the release of land for large-scale commercial agriculture. This has, so far, resulted in the acquisition of over 9000 ha of land in Ghana [36]. The BFP promoted large-scale land-based investment by encouraging the consolidation of large tracks of arable land for crop production [35].

Additionally, one of the intervention areas of FASDEP for modernizing agriculture is the promotion of reforms for land acquisition for large-scale plantation crops, such as mangos, cashews, oil palm, and rubber [2]. Therefore, the current agricultural policy efforts have had a significant role in the recent wave of LSLA in Ghana, where both transnational and national actors eagerly acquire land on a large scale to produce food and energy for export [11,38].

## 4. Literature Review

One of the main reasons put forward by proponents of large-scale land acquisition (LSLA) to encourage such deals is the technological benefits that have been perceived to await host countries. For instance, the immediate former President of the International Fund for Agricultural Development (IFAD) from 2007 to 2017, Dr. Kanayo F. Nwanze, believed that there is a potential for win-win situations, as such land deals can bring in new technologies [39]. Also, [15] proposed that, among other benefits, workers can learn simple techniques such as crop rotation, intercropping, and line sowing, which are then easily transmittable from large to small farms. Further, [40] also believed that large farms can benefit neighboring smallholders via a number of channels that include access to improved techniques.

In addition, a number of studies have established how large-scale investments have led to the transfer of technology. In fact, both theoretical and empirical studies have established

that large-scale investments play a vital role in promoting the increased technological level of farmers in recipient countries. For instance, building on the theory of agricultural intensification by Boserup [41], Behrman et al. [42] argued that large-scale land deals bring about the introduction of high inputs of capital, new technologies, and agrochemicals in areas where they are located. Following on from the conceptualization that large-scale investment can bring about technology adoption, [15] assessed spillover effects from a large farm establishment in Mozambique. The results suggested positive short-term effects from newly established large farms on the adoption of agricultural practices and input used by small farms less than 50 km from newly established large operations. In Nigeria, [14] also examined the effects of the presence of foreign migrant farmers on small-scale farming systems. The results from their study revealed significant increases in seed rate, fertilizer, and other chemicals per farmer in the area when compared to the situation that was prevalent before white farmers settled there.

Despite the fact these deals can yield spillover effects for local farmers, there are still others that have mixed feelings about the spillover effects of such deals. For example, Ilse Aigner, Germany's Agriculture and Consumer Protection Minister from 2008 to 2013, told Reuters that every country should own their land to make sure they can feed their own people [39]. The foreign direct investment literature in particular is also noted for encouraging this direction. A notable example is the work of [43], which elaborated on the relationship between land leases to foreigners and local farmers' modernization efforts. Specifically, [43] developed an occupational choice model to show that the introduction of high capital inputs, new technologies, and agrochemicals by investor companies can bring about positive or negative changes in the livelihoods of local occupants. Building on the occupational choice model of Dessy et al. [43], Kleemann and Thiele [44] showed that the effect of large-scale land acquisition rather depends on the investment model of investors. If the investor plants capital-intensive staple food crops, spillovers to local farmers will be rare because farmers do not get the chance to learn the newly introduced technologies as contacts with the investment farm is limited. However, if the investor plants cash crops, the technological spillover is significant because those farmers will learn technologies through contract farming, and then inform local farmers about the benefits and correct use of technologies. This can then lead to the purchase of newly introduced technologies for use on their own farms. However, none of the propositions from these models have been empirically tested.

However, in Ghana, only [16] examined the effects of land acquisition on the decision to invest in farming. The results revealed that the increasing appropriation of communal lands for biofuel plantations without consultation or fair and adequate compensation to the indigenous land holders has resulted in low investments in the farms of the affected farmers. However, in as much as the study is applauded, it is also important to note that the effects of such acquisitions of investment were indirectly captured through uncertainty, tenure insecurity, and farm sizes cultivated among farmers in affected communities. Moreover, the study focused on only two districts in southern Ghana, and the results are therefore not representative for all areas in Ghana.

## 5. Methodology

### 5.1. Conceptual Framework

The theoretical basis for examining the relationship between LSLA and farm investments is deeply rooted in neoclassical theories depicting the relationship between land tenure insecurity and farm investment. One of the central arguments of the neoclassical theories of land is that inefficient land tenure arrangements create uncertainties about the returns to the producer for his/her investments, thereby dissipating long-term investments in the land and reducing agriculture's contribution to social wellbeing [45]. This school of thought, therefore, argued that communal land tenure systems should be replaced by systems of individualization or privatization of land tenure to enhance access to capital (including credit) for fixed-place investment. Such arguments have been a subject of de-

bate in the theoretical literature, and have since paved the way for later-day neo-classical economists who researched how such arrangements caused tenure insecurity for smallholders, thereby influencing different types of farm investments. Notable among these studies is [5], who formulated a conceptual model relating tenure insecurity to credit supply and different types of farm investment in Thailand.

Our framework for testing the effect of LSLA on farm investment is based on the framework of [5] on the relationship between tenure insecurity and farm investment. Although the policy environment in Thailand is different from Ghana, the framework of [5] in Thailand is to some extent applicable in Ghana. This is because Ghana practices legal pluralism where statutory and customary land regimes coexist.

Under legal pluralism, land is vested in the hands of both traditional and state authorities who reportedly lease land to investors (e.g., [9,46]). However, such leases come with appropriation and displacement [17]. Appropriation and displacement create a perception of tenure insecurity [47], which is strongly linked to farm investment. Thus, based on the assumption that uncertainty (perception of tenure insecurity) increases among households exposed to LSLA, the question of how LSLA affects farm investment in northern Ghana is researchable using the model of [5].

According to [5], the farmer chooses between investments in capital equipment K, which is not affected in the event of eviction, but loses value due to forced sale of equipment and implements; land improvements M, which are completely lost in an eviction; and nonagricultural activities and assets Z, which are unaffected by eviction. The farmer invests in the first period and produces in the second, with the objective of maximizing the probability-weighted sum of expected terminal wealth in the absence and presence of eviction. The terminal wealth is made of aggregate value of land, production, and returns to nonagricultural activities with less costs resulting from borrowing. Application of the first-order conditions for a maximum and Cramer's rule yields the effects of increased risk of eviction on capital and land improving techniques, as defined in the following reduced form equations:

$$K_{ij}^* = K(\Phi, W_o, \ A, S) \tag{1}$$

$$M_{ij}^* = M(\Phi, W_o, \ A, S) \tag{2}$$

where $K_{ij}^*$ and $M_{ij}^*$ in Equations (1) and (2) represent the change in capital investment and land-improving techniques due to a change in the rate of eviction $\Phi$. Equations (1) and (2) further imply that a change in investment in capital equipment $K_{ij}^*$, and land- and yield-improving techniques $M_{ij}^*$ are influenced by the rate of eviction $\Phi$, the initial wealth $W_o$, amount of land A, and human capital S. However, the objective of this study is to examine how exposure to LSLA affects investment in land- and yield-improving techniques $M_{ij}^*$. In that regard, our immediate task is to conceptualize how exposure LSLA enters the capital and farm investment functions in Equations (1) and (2).

Following the literature regarding land deals (e.g., [43,44]) and tenure insecurity (e.g., [16,48]), we argue that there are several channels through which LSLA influences household's investments $K_{ij}^*$ and $M_{ij}^*$ in Equations (1) and (2). Among these channels are rate of eviction $\Phi$ and amount of land A owned by a household. As evidenced in the literature (e.g., [16,47,48]), LSLA has a strong bearing on eviction rates. For instance, [47] have argued that the probability of eviction is not only evidenced in land redistribution and missing formal land markets, but also in the presence of appropriation, which mostly comes with LSLA. Therefore, given other parameters X, tenure insecurity, captured by the probability of eviction $\Phi$ from one's parcel $A_T$, is correlated with exposure to LSLA, i.e., $\Phi = f(\text{LSLA}|X)$ or $LSLA = f(\Phi|X)$, under the assumption that the relationship between LSLA and probability of eviction $\Phi$ is simultaneous.

Moreover, the farm investment index by $K_{ij}^*$ and $M_{ij}^*$ is influenced by LSLA through the amount of land available for household A, after the acquisition of farmland (e.g., [43,44]). Thus, in actual fact, $A = A_T - LSLA$; and $A_T$ and $LSLA$ are the total amount of land and amount of land loss to foreign or domestic investors, respectively. For simplicity, one can

therefore assume a possible substitution of LSLA in place of $\Phi$ or $A$ in Equations (1) and (2), without further complications. Thus, aside from the A effect of LSLA, the relationship between LSLA, and $K_{ij}^*$ and $M_{ij}^*$ is mediated by $\Phi$, and hence, is indirect; this is tested later.

Based on the expositions above, Equations (1) and (2) can be employed to test the effect of LSLA on farm investment in land and yield improvement techniques. It is also important to point out that a reverse causality between farm investments and LSLA is also possible, since LSLA increases the perceived risk of losing land rights and higher investments to enhance claims of land [49]. In the next section, we present the data and strategy for testing the effect of LSLA on investment, and the potential reverse causality using Equations (1) and (2).

*5.2. Data Information*

The study area for this research was northern Ghana (now demarcated into Northern, Savannah, and North East regions). Several studies (e.g., [50]) have found that investors' acquisition of land on a large scale for agricultural investment is strongly influenced by the availability of potential arable crop lands. Northern Ghana was therefore selected for this study because it is known to have vast areas of potential arable cropland (PACs) in Ghana [51], and is therefore likely to attract investors. Aside from the vast PACs, the area is characterized by a single rainy season, which is of a longer duration (lasting between May and October) and is therefore well suited for agriculture, in which the majority of the inhabitants are involved. The area is also less dense (35 pp/sq. km), with a total population of 2,479,461 inhabitants, but with a land area of about 70,384 square kilometers [51].

The land is, however, controlled by a two-tier system of governance. The four paramount chiefs in the region, namely the Ya-Na of Dagbong, the Nayiri of Mamprugu, the Bimbilla Naa of Nanung, and the Yagbonwura of the Gonja Traditional area, constitute the first tier. Under the first tier, the land is governed by various unwritten laws and practices which present several challenges, including a plurality of management and multiple sales [28]. The second tier is the local government system in which land is governed by state laws. The problem, however, is the state agencies under whose care land is entrusted. Currently, both the Lands Commission and the Land Title Registry claim to be the final authorities in charge of land management [52].

These problems contribute towards a weak system of land governance, and hence, make the area a hotbed for LSLA by domestic and foreign agribusiness investors [25]. Special cases include the 23,762 hectares acquired by Biofuel Africa Limited in the Central Gonja and Mion districts [28]; the Integrated Tamale Fruit Company (ITFC) in Savelegu district, which has a nucleus farm of over 160 hectares and over 2000 out-growers [9]. Several other examples of LSLA by returning citizens, retiring from the civil service, and others seeking to invest their incomes in landed property, also exist in the area.

Regarding the data, a total of 690 exposed and nonexposed agricultural households were selected from a target population of 240,238 agricultural households in northern Ghana. The sample size was determined following [53]. The households were selected through a multistage sampling technique. The first stage consisted of the selection of six districts (Figure 1) based on the predominance of vast tracks of arable land under commercial deals.

Documented information from the Northern Regional Lands Commission revealed that Central and North Gonja from the Savannah region, Mampurugu-Muagdure from the North East Region and Mion, Sagnarigu, and Savelegu districts from the Northern Region (Figure 1) dominate in arable land under commercial deals or large-scale land acquisition (LSLA) and represent about 98.87% of the total deals documented (see Table A3 of Appendix A). For this reason, these districts (as shown in Figure 1) were selected for the study.

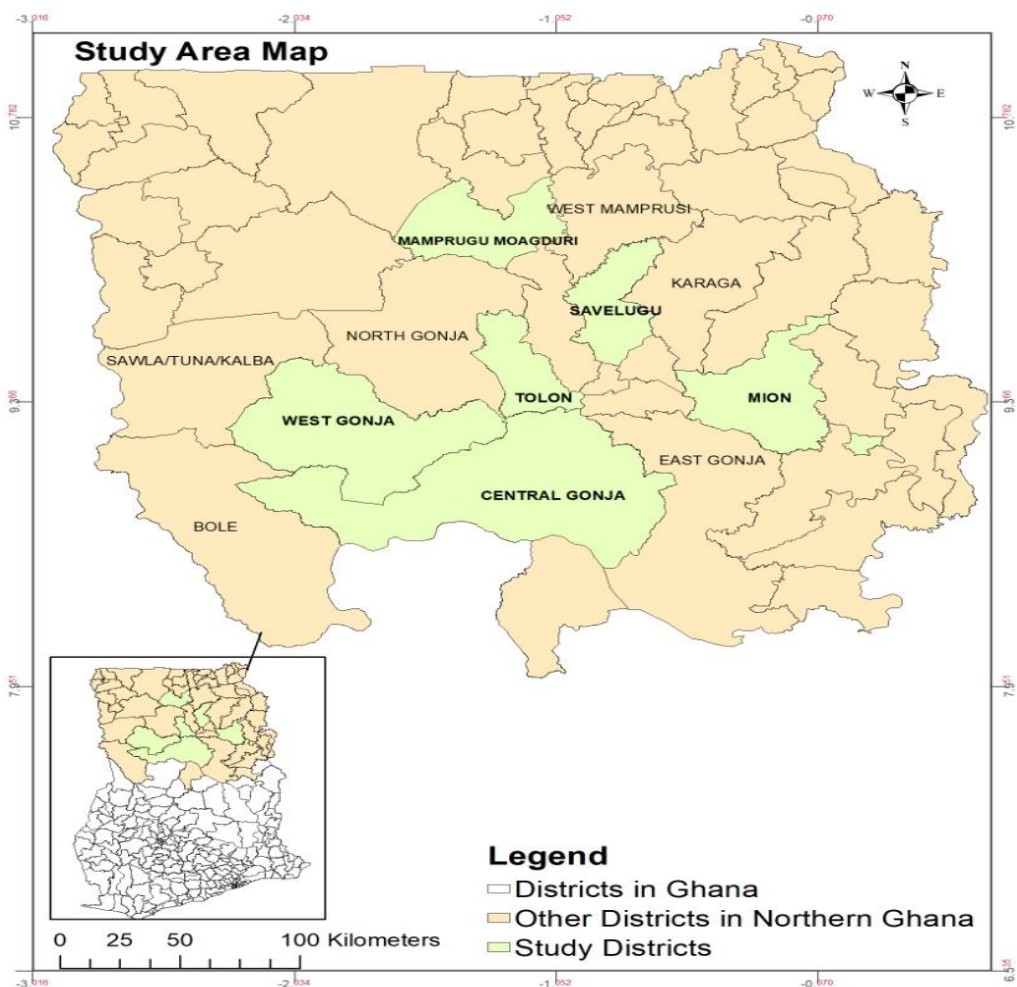

**Figure 1.** A map of northern Ghana showing the study districts.

In the second stage, 23 affected communities were selected from the 6 districts through a reduction process (the reduction process involved scoping of the selected districts which identify 41 communities affected by LSLA; profiling the 41 affected communities; contrasting the communities to identify those best represented by LSLA from domestic and foreign entities; and sampling of 23 out of 41 affected communities based on predominance of LSLA by domestic and foreign entities). Although the 23 communities identified represented those communities affected by LSLA from domestic and foreign entities, it was difficult to locate agricultural households under direct exposure (i.e., households losing farmland, labor, forest resources, etc., to domestic or foreign entities) and indirect exposure (i.e., households living nearby affected households; losing uncultivated land; having limited land due to enclosures). The difficulty stemmed from the fact that there was no comprehensive list of agricultural households exposed to large-scale land acquisition (LSLA). To generate a list, common places—where farmers normally converge to play local games or discuss daily activities—were identified in every community. Farmers from these locations were then asked to supply the names of agricultural households exposed to LSLA. The names supplied were compiled into a list of 955 affected households. To be sure that the list actually consisted of farmers affected by LSLA by domestic and foreign entities, questions based on the operational definition of LSLA were asked to farmers. These questions included whether farmers had actually lost land, the origin of the acquirers, and the details of the losses due to LSLA. The responses were compared to our operational definitions and those that conformed with it were included in the final list. The final list consisted of 954 households affected by LSLA by foreign and domestic entities. In sampling the

690 exposed households, the simple random sampling technique was employed. Of all 690 agricultural households, there were 138 exposed directly and indirectly to LSLA by domestic and foreign entities, and 138 nonexposed households.

After sampling, semi-structured questionnaires were administered through a household survey. The survey questionnaire captured information based on previous studies on LSLA (e.g., [20,50,54]) and tenure security-farm investment nexus (e.g., [5,55,56]). The information concerned a household's exposure to LSLA, household power relations (e.g., gender, education, age of the head of the household, household size, etc.), location characteristics (e.g., access to market, roads, irrigation facilities, extension, land institutions, size of marginal land, soil fertility, market, infrastructure, credit programs, etc.), farmland access, labor supply, farm investment, food production, and food security. Some of the data were, however, dropped because of nonresponses, leaving a study sample of 664 agricultural households. Figure 2 presents the study area showing the locations of sampled households.

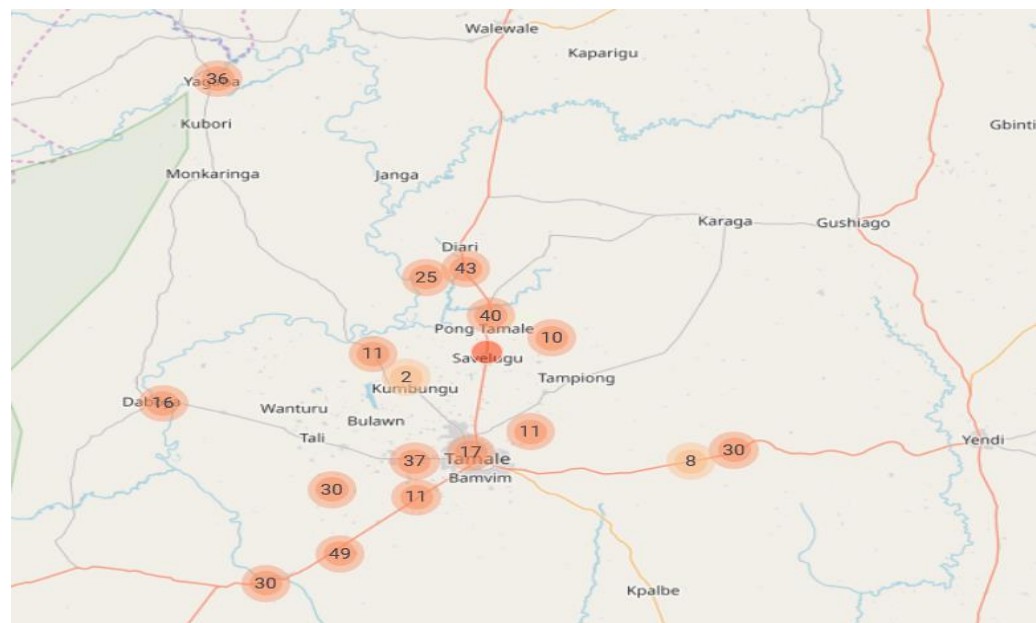

**Figure 2.** Overview of the study area. Note: The orange-colored circles represent the location of the respondent during the interview, while the figure in the orange-colored circles represent the number interviewed at that particular location. Source: Field Survey, 2018.

Table 2 presents the variables definition/measurement and descriptive statistics of the variables included in the analysis. Among the study sample, 21% represent households under non-exposure, while 20% represent each of the households under direct and indirect exposure to LSLA by domestic and foreign entities. On average, the sample household income was GH¢5,095.65 (USD884.66) (GH¢ is Ghana cedis; GH¢5.76 was equivalent to US$1 at the time of the study). Concerning gender, the statistics indicate more males than females in the sample. Thus, any policy concerning resources for production is more likely to affect more men and the livelihood of many households, since males are mostly the breadwinners of their families. The average age of respondents in the sample was 47 years. This suggests that the sampled households were represented by adults or a mature group of respondents. Furthermore, educational level was low, given that the average number of years spent in school was about 2 years in the sample. This was therefore likely to have more implications for households' land access and investment in the area.

**Table 2.** Variable definition/measurement and a priori expectations.

| Variable | Definition/Measurement | Mean (SD) |
|---|---|---|
| Wells | Dummy (1 if the household had invested in construction of wells and 0 if otherwise) | 0.19 (0.39) |
| Local catchments | Dummy (1 if the household had invested in construction of local catchments/dugouts and 0 if otherwise) | 0.33 (0.47) |
| Drip irrigation facilities | Dummy (1 if the household had invested in irrigation facilities and 0 if otherwise) | 0.12 (0.33) |
| Intercropping | Dummy (1 if the household had invested in intercropping with nitrogen fixing crops and 0 if otherwise) | 0.41 (0.49) |
| Minimum tillage | Dummy (1 if the household had invested in minimum tillage and 0 if otherwise) | 0.21 (0.41) |
| Crop residue retention | Dummy (1 if the household had invested in crop residue retention and 0 if otherwise) | 0.28 (0.45) |
| NPK | Dummy (1 if the household had invested in NPK and 0 if otherwise) | 0.62 (0.49) |
| SOA | Dummy (1 if the household had invested in sulphate of ammonia (SOA) and 0 if otherwise) | 0.35 (0.48) |
| Urea | Dummy (1 if the household had invested in Urea and 0 if otherwise) | 0.49 (0.50) |
| Household income | Aggregate income from the farm, off-farm wages, salary, petty-trade, and other activities (in GH¢) | 5095.65 (21.03) |
| Fertilizer subsidy | Dummy (1 if farmer benefited from the 2017/2018 fertilizer subsidy programme, 0 if otherwise) | 0.75 (0.43) |
| Gender | Dummy (1 if household head is male, 0 if otherwise) | 0.93 (0.26) |
| Age | Age of household head (years) | 46.97 (2.87) |
| Household size | Number of people residing in a household | 12.44 (7.28) |
| Education | Number of years spent in formal education | 1.97 (3.86) |
| Farm size | All the land under the management and control of household without regard to title, legal form, size, or location (ha) | 6.39 (3.78) |
| Leadership | Dummy (1 if household head is in any leadership position; 0 if otherwise) | 0.26 (0.44) |
| Sagnarigu [1] | Dummy (1 if farmer is located in Sagnarigu district, 0 if otherwise) | 0.17 (0.38) |
| Mion [1] | Dummy (1 if farmer is located in Mion district, 0 if otherwise) | 0.09 (0.29) |
| Central Gonja [1] | Dummy (1 if farmer is located in Central Gonja district, 0 if otherwise) | 0.18 (0.39) |
| Savelegu [1] | Dummy (1 if farmer is located in Savelegu district, 0 if otherwise) | 0.36 (0.48) |
| Yagba-Kubori [1] | Dummy (1 if farmer is located in Yagba-Kubori district, 0 if otherwise) | 0.17 (0.38) |
| North Gonja [1] | Dummy (1 if a farmer is located in North Gonja district, 0 if otherwise) | 0.02 (0.15) |
| Water resources | Dummy (1 if there is available water resource in the village; 0 if otherwise) | 0.41 (0.49) |
| Road | Distance to the nearest weathered road (km) | 0.35 (0.48) |
| Credit | Dummy (1 if the household has access to credit; 0 if otherwise) | 0.33 (0.47) |
| Social group | Dummy (1 if farmer is a member of social group; 0 if otherwise) | 0.39 (0.49) |
| Knowledge | Dummy (1 if the household has prior knowledge of households affected by LSLA; 0 if otherwise) | 0.61 (0.49) |
| Good fertile [2] | Dummy (1 if the fertility of the soil is good; 0 if otherwise) | 0.38 (0.49) |
| Moderately fertile [2] | Dummy (1 if the fertility of the soil is moderate; 0 if otherwise) | 0.45 (0.50) |
| Poorly fertile [2] | Dummy (1 if the fertility of the soil is poor; 0 if otherwise) | 0.17 (0.38) |
| Land institution | Dummy (1 if a formal land institution such as Lands Commission, and Land Use and Spatial Planning Department is available; 0 if otherwise) | 0.34 (0.48) |

**Table 2.** *Cont.*

| Variable | Definition/Measurement | Mean (SD) |
|---|---|---|
| Non-exposure | 1 if households did not lose land directly or indirectly to domestic or foreign entities; 0 if otherwise | 0.21 |
| **Exposure to LSLA by domestic entities** | | |
| Direct exposure | 1 if households lost farmland, labour, and farmland-based resources to domestic entities; 0 if otherwise | 0.20 |
| Indirect exposure | 1 if household live nearby affected households or lost uncultivated land; have limited land and cannot practice fallowing, monocropping because the land has become scarce due to enclosures; 0 if otherwise | 0.20 |
| **Exposure to LSLA by foreign entities** | | |
| Direct exposure | 1 if households lost farmland, labour, and farmland-based resources to foreign entities; 0 if otherwise | 0.20 |
| Indirect exposure | 1 if household live nearby affected households or lost uncultivated land; have limited land and cannot practice fallowing, monocropping due to scarcity of land caused by foreign enclosures; 0 if otherwise | 0.20 |

Notes: STI and LTI denote short-term investment and long-term investment, respectively. GH¢ is Ghana cedis (US\$1 = GH¢5.76 at the time of the study); [1] North Gonja is reference category in all estimations; [2] Poorly fertile is reference category in all estimations.

In terms of investments in land- and yield-improving techniques, about 62% of the sampled households had made investments in NPK. This represents the highest among all the land-improving techniques in this study. Urea was revealed as the next most patronized technique in terms of investment decisions, as about 49% of the sampled households were found using it. These techniques have been subsidized for households under the Ghanaian governments' fertilizer subsidy program, and hence, the high investment among households. Investment in intercropping was about 41%, probably because it is relatively easy and cheaper than irrigation (12%) and wells (19%), which are the least patronized techniques among the sampled households. Furthermore, around 35, 33, 28, and 21% of the sampled households were found to invest in sulphate of ammonia, local catchments, crop residue retention, and minimum tillage, respectively. Thus, investment in land improving techniques was not equal among the sampled households. This implies that some of the land improving techniques are either used by complementing other techniques or substituted for other techniques in this study. This is probably due to institutional structure and processes, government and traditional policies, and vulnerability contexts. However, these revelations are not enough to conclude that such factors actually compel farmers to prioritize investment in some land-improving techniques over others. In the next section, we examine exposure to LSLA—components of the vulnerability context (we have argued elsewhere ([25]) that LSLA is under the vulnerability context)—and its effects on investment in land-improving techniques.

### 5.3. Estimation Strategies

To test the effect of LSLA on investment decisions and level of investment in land improvement techniques, we specified the following general equation based on the expositions in Section 5.1:

$$M_{ik}^* = \delta_i E_{ik} + \vartheta_i X_{ik} + \mu_{ik} \tag{3}$$

where $M_{ik}^*$ represents a latent variable for investment decisions or level of investments made by household $i$ on land-improving technique $k$ [i.e., wells (W), local catchments (LC), drip irrigation facilities (DIF), intercropping (I), minimum tillage (MT), residue retention (RR), NKP, Sulphate of Ammonia (SOA), and Urea (U)]; $E_{ik}$ is a vector of direct and indirect exposure to LSLA under each of the domestic and foreign entities; $X_{ik}$ is a vector of household's tenure security, initial wealth, land, human capital, and other supply-side

variables; $\vartheta_i$ is the coefficient of $X_i$; $\mu_{ik}$ is a random term. Furthermore, Equation (3) implies that $\delta_i$ can be the effect of direct or indirect exposure to LSLA on latent variables for investment or level of investment $M_{ik}^*$. This is true under the assumption that no correlation exists between the investment decisions or error term $\mu_{ik}$ and direct and indirect exposure to LSLA $E_{ik}$ in Equation (3), i.e., $\rho = corr(\mu_{ik}, E_{ik}) = 0$. However, investment decisions on wells (W), local catchments (LC), drip irrigation facilities (DIF), intercropping with nitrogen fixing crops (I), minimum tillage (MT), residue retention (RR), NKP, Sulphate of Ammonia (SOA), and Urea (U) are not only inherently multivariate but may also be interrelated.

Moreover, households may report losing land (i.e., self-select into exposure to LSLA, $E_{ik}$) due to the benefit associated with land loss. Such benefit may be unknown to the researcher (missing variable problem), and thus, causes a correlation between $E_{ik}$ and $\mu_{ik}$. Furthermore, state and traditional authorities in charge of land sales may select the plots based on their quality attributes (often unobservable to researchers). Such a non-random process of selecting plots for acquisition may lead to a systematic difference between exposed (i.e., direct and indirect exposure) and non-exposed households. The differences may lead to a correlation between the error term $\mu_{ik}$ and exposure variables $E_{ik}$ (i.e., $corr(\mu_{ik}, E_{ik}) \neq 0$).

Moreover, it has been shown in empirical and theoretical studies that in many traditional tenure systems, farmers may minimize eviction rates through investment in land-improving techniques. For instance, [49] showed that a tenant or sharecropper who feels insecure can conserve or invest in the soil to minimize eviction by landlords. Other studies (e.g., [57]) also argued that farmers with a perceived risk of eviction may resort to undertaking higher investments, which in turn enhance their claims to the land. Given such potential simultaneity, $\rho = corr(\mu_{ik}, E_{ik}) \neq 0$. In this regard, $\delta_i$ in Equation (3) may not be taken as the true effect of direct and indirect exposure to LSLA under domestic and foreign entities if the potential selection bias and simultaneity are not accounted for.

### 5.3.1. Model Identification

Given the challenges enumerated above, there is the need to employ methods that best identify the impact of LSLA on farm investment. In an ideal world or randomized control trial setting, these are by randomly assigning individuals into treatment (i.e., direct exposure to LSLA by domestic entities, indirect exposure to LSLA by domestic entities, direct exposure to LSLA by foreign entities, and indirect exposure to LSLA by foreign entities) and control (non-exposure) groups, such that the only differentiating factor among exposure and non-exposure groups is exposure to LSLA. Unfortunately, LSLA is not a new phenomenon in Ghana and has been an ongoing practice since the colonial period. This implies that the random assignment of farm households to treated and nontreated groups (a requirement for estimating the impact of a program) is not possible.

Given the above caveats, there is the need to employ approaches that allow for the control of assignment to treatment without relying on random assignment. In this regard, the quasi-experimental research approach remains a better option. Generally, two main methods exist under the quasi-experimental research approach: namely, the semiparametric and parametric methods.

The semiparametric methods include propensity score matching (PSM) techniques. However, PSM techniques assume that exposed and non-exposed households are systematically different only in observed characteristics [58]. However, unobserved characteristics (e.g., farmers' innate abilities, etc.) may also simultaneously influence exposure to LSLA and outcomes of interest. Thus, ignoring such factors may lead to biased estimates.

On the other hand, parametric methods include the OLS, nonlinear regressions, Heckman two-step method, and IV-approaches. Although these methods impose functional form assumptions on the coefficients of covariates and may require instruments that are difficult to come by, they control for endogeneity resulting from observed and unobserved variables.

To account for the potential endogeneity and estimate the effect of LSLA on investment decisions and level of investment in land improving techniques, IV-approaches, including the control function approach [21] and the extended regression models [22] are employed, respectively. The generalized two-stage simultaneous probit [59] and two-stage conditional maximum likelihood [60] also control for endogeneity. However, such procedures are not legitimate when both the response and endogenous explanatory variables are discrete. It is for this reason we resort to the control function approach (CFA) [21] and the extended regression models (ERMs) [22].

The CFA is employed to examine the effect of direct and indirect exposure to LSLA by domestic and foreign entities on investment decisions in land-improving techniques, while the ordered probit regression with endogenous treatment is employed to examine the effect of direct and indirect exposure to LSLA by domestic and foreign entities on the level of investment in land-improving techniques. In the next section, we present how the effect of exposure to LSLA on farm investment decisions and level of investment are estimated using these methods.

### 5.3.2. Estimating the Effect of LSLA on Investment Decisions

As stated earlier, the control function approach (CFA) recently developed by [21] is employed to examine the effect of direct and indirect exposure to LSLA on investment decisions. The CFA is an instrumental variable IV-type approach that allows for the estimation of nonlinear models with discrete endogenous explanatory variables (see for instance, [21]) and has been employed recently by [61]. A similar approach has been employed in related studies (e.g., [49,56]). As in other IV-type approaches, the consistency of the CFA estimates depends on the strength of the IVs and their valid exclusion from the main, second-stage regression. The instrument variables used will be defined shortly. The CFA proceeds in two stages. In the first stage, generalized residuals are estimated from a set of probit models relating direct and indirect exposure to LSLA under domestic and foreign entities to a set of exogenous covariates $X_i$ and instrumental variables $Z_i$. The probit model for each of the direct and indirect exposure to LSLA under domestic and foreign entities are estimated from the equation below:

$$E_{ij}^* = a_i X_i + \psi_i Z_i + \eta_i \tag{4}$$

$$E_{ij} = \begin{cases} 1 \ if \ E_{ik}^* > 0 \\ 0 \ if \ E_{ik}^* \le 0 \end{cases} \tag{5}$$

where $E_{ij}^*$ is a latent variable for direct and indirect exposure to LSLA under domestic and foreign entities; $E_{ij}$ is the observed variable for direct and indirect exposure to LSLA under domestic and foreign entities; $a_i$ and $\psi_i$ are the coefficients of $X_i$ and $Z_i$, respectively.

In the second stage, the generalized residuals from the first-stage probit models are included in the main, second-stage regression along with the endogenous explanatory variables. In this study, the second stage regression is a multivariate probit model because households invest in a mix of land-improving techniques to deal with a multitude of agricultural production constraints. Thus, investment decisions on wells (W), local catchments (LC), drip irrigation facilities (DIF), intercropping (I), minimum tillage (MT), residue retention (RR), NKP, Sulphate of Ammonia (SOA), and Urea (U) are not only inherently multivariate, but may also be interrelated. For instance, a household investing labor and/or money in wells and drip irrigation facilities may have less labor and money available for the construction of local catchments, since both of these techniques are labor intensive and expensive (loss of labor and income effect). Similarly, investment in intercropping with nitrogen-fixing crops improves soil fertility and yields, and may discourage investment in land- and yield-improving NKP, Sulphate of Ammonia, and Urea. On the other hand, investment in wells, drip irrigation facilities, and local catchments may encourage investment in intercropping with nitrogen-fixing crops, minimum tillage, and residue retention to improve soil quality and yields. Ignoring such potential interdependence between the

techniques may lead to biased estimates of the impact of LSLA on investment decisions. Thus, to deal with the potential correlation between the land improving techniques, we specify the following multivariate probit model in the second stage:

$$M_{ik} = \delta_0 + \delta_1 E_i + \lambda_i \hat{E}_i + \delta_2 X_i + \omega_i \ k = W, LC, DIF, I, MT, RR, NPK, SOA, U \quad (6)$$

$$M_{ik} = \begin{cases} 1 \ if \ M_{ik}^* > 0 \\ 0 \ if \ M_{ik}^* \leq 0 \end{cases} \quad (7)$$

where $M_{ik}$ is the observed investment decision based on $M_{ik}^*$, which is as defined earlier in (3) for each of the land improving techniques $k$ in (6); $E_i$ is the vector of the endogenous variables (i.e., direct and indirect exposure to LSLA under domestic and foreign entities), $\hat{E}_i$ the vector of the generalized residual from the first-stage probit estimations (i.e., probability of direct and indirect exposure to LSLA under domestic and foreign entities); $X_i$ is a vector of exogenous covariates; and $\delta_1$, $\lambda_i$, and $\delta_2$ are the coefficients of the vector of endogenous variables, $E_i$, vector of the generalized residuals $\hat{E}_i$, and vector of exogenous covariates $X_i$. Assuming that the error term $\omega_i$ in Equation (6) jointly follows a multivariate normal distribution with mean zero and variance 1, and the covariance matrix $\sum$ expressed as: $(\omega_W, \omega_{LC}, \omega_{DIF}, \omega_I, \omega_{MT}, \omega_{RR}, \omega_{NPK}, \omega_{SOA}, \omega_U) \sim MVN(0, \sum)$, the maximum likelihood method can be employed to estimate the parameters and the correlations of the error terms [56,62]. The multivariate probit model tests the potential interdependency between investment decisions using the sign of the correlation coefficient between the different investments. Specifically, the sign of correlation coefficients between the investment decisions provides an idea as to whether there is complementarity (positive correlation) or substitutability (negative correlation) between investment decisions regarding land- and yield-improving techniques [62]. In Equation (6), the generalized residuals simultaneously correct for and permit the testing of the endogenous nature of the explanatory variables. According to [21], including the generalized residuals in the second- stage nonlinear model along with a set of exogenous covariates and the endogenous variables might provide an accurate correction for endogeneity. The significance of $\lambda_i$ is an indication that exogeneity is rejected and that the inclusion of the residuals corrects for endogeneity.

We also test the reverse causality between investment decisions and exposure to LSLA. Thus, the effect of each of the investment decisions in land-improving techniques on direct and indirect exposure to LSLA under each of the domestic and foreign entities was estimated using the CFA proposed by [21]. The following systems of equations were estimated under CFA [21]:

$$M_{ik} = \delta_0 + \delta_2 X_i + \tau I + \epsilon_i \quad (8)$$

$$E_{ij} = \delta_0 + \delta_1 M_i + \gamma_i \hat{M}_i + \delta_2 X_i + \xi_i \quad (9)$$

where $E_{ij}$ denotes the probability of direct and indirect exposure to LSLA under domestic and foreign entities, $M_i$ is the investment decisions assumed to be endogenous, $\hat{M}_i$ is the generalized residuals to be generated from Equation (8). Using Equation (8), the generalized residuals will be estimated from a set of probit models relating investment decisions on each of the nine land-improving techniques to a set of exogenous covariates $X_i$ and instrumental variables $I_i$. This constitutes the first stage of the CFA.

In the second stage of the CFA, the generalized residuals ($\hat{M}_i$) from Equation (8), the endogenous variable ($M_i$), and a set of exogenous covariates ($X_i$) are introduced into the second-stage equation (i.e., Equation (9)) to examine the effect of LSLA on the investment decisions of households in northern Ghana. In the case of this study, the second stage of the CFA estimation effect of investment on direct and indirect exposure to LSLA under domestic and foreign entities (i.e., Equation (9)) will use the multinomial logit regression, since exposure to LSLA was found to compose of dissimilar categories (we run a Hausman tests for the IIA assumption following [63], and the results indicates that the assumption is not violated). Again, coefficient $\gamma_i$ is a valid test of the null hypotheses that the investment

variables are exogenous in the exposure equations [55]. If coefficient $\gamma_i$ is significant, exogeneity is rejected and their inclusion corrects for endogeneity.

### 5.3.3. Identification of Farm Investment and Exposure to LSLA Equations

Proper identification of the farm investment Equation (6) and Equation (11) for exposure to LSLA requires that some of the variables included in the first-stage estimation of investment and exposure to LSLA are excluded from the second-stage estimations of exposure to LSLA and households' farm investments. Thus, in the second-stage multivariate probit equations for farm investment under domestic and foreign entities, we exclude land institutions (measured as 1 if a formal land institution such as Lands Commission and Land Use and Spatial Planning Department are available; 0 if otherwise) but included it in the first-stage equations of direct and indirect exposure to LSLA under domestic and foreign entities.

Intuitively, the availability of land institutions tends to enhance contact and information about land, actors/stakeholders, and their attributes, thereby decreasing rate of exposure to LSLA. Moreover, the availability of land institutions shows not just the presence, but the quality of policies, law and order, land governance, and protection of investors and their investments, and therefore may enhance LSLA by both domestic and foreign investors [50].

With regards to the second-stage multinomial logit equation for exposure to LSLA under each of the domestic and foreign entities, we exclude prior knowledge of other households affected by LSLA (measured as 1 if any member of the household had prior knowledge of other communities affected by LSLA; 0 if otherwise) but include it in the first-stage equations of farm investment. Farmers with prior knowledge of other households affected by LSLA in other communities tend to increase investment in land-improving techniques to minimize exposure to LSLA. Thus, first-hand information regarding the types of land acquired by investors reduces the risk of eviction through farm investment.

For instance, [64] revealed that farmers who had prior information from relatives and related networks about LSLA enhanced the security of their remaining land through investment in rubber plantations. These instruments have been used in recent empirical studies for impact evaluation (e.g., [25]). Even though the instruments are intuitively strong, the validity of some of the instruments may be questioned. For instance, it may be argued that the availability of land institutions may enhance rates of registration, certification, and security of land tenure, thereby enhancing farm investment decisions. Our argument is that availability is confined to only the presence of land institutions, but not household usage of such institutions. However, the assumption of exclusion restrictions may be violated, especially if it turns out that both exposure to LSLA and investment decisions are influenced by availability of land intuition and prior information about households exposed to LSLA. We test for relevance and exogeneity of instruments by following [65] overidentification test statistics, with χ2 distribution and degrees of freedom equal to the number of excluded instruments. According to [65], the test involves estimating an alternative version of Equations (8) and (9) with the instruments. The insignificance of the coefficients of the instruments in the estimations is then considered as evidence that the instruments can be excluded from Equations (6) and (9).

### 5.3.4. Estimating the Effect of LSLA on Level of Investment in Land-Improving Techniques

As indicated already, nine investment decisions were captured in this study and include wells (W), local catchments (LC), drip irrigation facilities (DIF), intercropping with nitrogen-fixing crops (I), minimum tillage (MT), residue retention (RR), NKP, Sulphate of Ammonia (SOA), and Urea (U). Based on households' responses to investment decisions in these techniques, an outcome variable capturing the number of techniques involved in households' investment decisions was generated.

Following [66], an ordinal outcome measuring levels of investment was generated from the number of techniques involved in households' investment decisions. The ordinal

outcome is categorized as follows: (i) category $M_{ik} = 0$, for households who invest in none of the land improving techniques on their plots; (ii) category $M_{ik} = 1$, for households who invest in only one such practice on their plots; (iii) category $M_{ik} = 2$, for households who invest in a combination of two such practices on their plots; (iv) category $M_{ik} = 3$, for households who invest in a combination of three such practices on their plots; (v) category $M_{ik} = 4$, for households who invest in a combination of four such practices on their plots; and (vi) category $M_{ik} = 5$, for households who invest in a combination of five or more such practices on their plots.

Given the ordered nature of the outcome variable and our interest in estimating the effect of exposure to LSLA on the level of investment in land-improving techniques, we first needed to specify an ordered probit model. Drawing from the latent response model of investment in Equation (7), we specify the level of investing household as:

$$M_{ik} = \begin{cases} 0, \; if \; M_{ik}^* \leq c_1 \\ 1, \; if \; c_1 < M_{ik}^* \leq c_2 \\ \quad \vdots \\ 5, \; if \; M_{ik}^* > c_5 \end{cases} \tag{10}$$

where $c$ is the index of unknown cut points identifying the boundaries of moving through the different levels of investment in land-improving techniques; and $M_{ik}$ and $M_{ik}^*$ are as previously defined. The formulas for the probabilities with the six observed results for the ordered probit model are as follows:

$$prob(M_{ik} = 0 | X_{ik}, \; E_{ik}) = \Phi(c_1 - \delta_i E_{ik} - \vartheta_i X_{ik}) \tag{11a}$$

$$prob(M_{ik} = 1 | X_{ik}, \; E_{ik}) = \Phi(c_2 - \delta_i E_{ik} - \vartheta_i X_{ik}) - \Phi(c_1 - \delta_i E_{ik} - \vartheta_i X_{ik}) \tag{11b}$$

$$prob(M_{ik} = 2 | X_{ik}, \; E_{ik}) = \Phi(c_3 - \delta_i E_{ik} - \vartheta_i X_{ik}) - \Phi(c_2 - \delta_i E_{ik} - \vartheta_i X_{ik}) \tag{11c}$$

$$prob(M_{ik} = 3 | X_{ik}, \; E_{ik}) = \Phi(c_4 - \delta_i E_{ik} - \vartheta_i X_{ik}) - \Phi(c_3 - \delta_i E_{ik} - \vartheta_i X_{ik}) \tag{11d}$$

$$prob(M_{ik} = 4 | X_{ik}, \; E_{ik}) = \Phi(c_5 - \delta_i E_{ik} - \vartheta_i X_{ik}) - \Phi(c_4 - \delta_i E_{ik} - \vartheta_i X_{ik}) \tag{11e}$$

$$prob(M_{ik} = 5 | X_{ik}, \; E_{ik}) = 1 - \Phi(c_5 - \delta_i E_{ik} - \vartheta_i X_{ik}) \tag{11f}$$

where $\Phi(.)$ is the standard normal distribution function; $X_{ik}$ and $E_{ik}$ are as earlier defined; $\delta_i$ and $\vartheta_i$ are the associated parameters to be estimated.

Of particular interest in this study is a proper estimation of $\delta_i$ in Equation (11)a–f. As previously explained, the index of direct and indirect exposure to LSLA under domestic and foreign entities ($E_{ik}$) is potentially endogenous due to selection bias and simultaneity. Thus, proper estimation of $\delta_i$ requires accounting for such endogeneity. In this study, we employed ordered probit regression with endogenous treatment and probit options.

Ordered probit regression with endogenous treatment is one of the extended regression models (ERMs) recently introduced in STATA [22] for probit and ordered probit models that allow binary, ordinal endogenous, and exogenous covariates. The features of the ERMs may be used separately or in any combination. In this study, ordered probit regression with endogenous treatment was estimated using the command 'eoprobit', available in STATA software [22]. An important feature of interest in the ERMs, including the ordered probit regression with endogenous treatment, is that it allows testing of the extra effect of endogenous treatment due to other covariates. This allows us to test the effects of exposure to LSLA through tenure insecurity, as established in our theoretical model (Section 5.1).

Like all IV approaches, ordered probit regression with endogenous treatment requires identification. Our models for the level of investment were identified using land institution.

## 6. Results and Discussion

The main objective of this study was to examine the relationship between direct and indirect exposure to LSLA under domestic and foreign entities on households' investment

in land-improving techniques. The CFA and ordered probit regression with endogenous treatment were employed to control endogeneity, and to estimate a system of equations involving farm investments and exposure to LSLA under domestic and foreign entities. The results are presented in the following sections.

### 6.1. Effect of Exposure to LSLA

As previously mentioned, exposure to LSLA was captured as a polychotomous variable (i.e., non-exposure to LSLA, direct exposure to LSLA by foreign entities, indirect exposure to LSLA by foreign entities, direct exposure to LSLA by domestic entities, and indirect exposure to LSLA by domestic entities). Thus, the results were obtained using a multinomial logit regression model in the second stage of the control function approach (CFA where the potential endogenous variables (i.e., investment in land- and yield-improving techniques, including wells, local catchments, drip irrigation facilities, intercropping with nitrogen fixing crops, minimum tillage, residue retention, NKP, Sulphate of Ammonia, and Urea) in this study) and their residuals from a set of first-stage of probit models were incorporated into a second-stage MNLM model. The first stage results of the determinants of investment in each of the land-improving techniques can be found in Table A1 of the Appendix A. For the purposes of brevity, we did not discuss these results here, since determinants of investment in land-improving techniques are presented and discussed in Section 6.2. On the other hand, Table 3 presents the regression results of determinants of direct and indirect exposure to LSLA under domestic and foreign entities.

As shown in Table 3, the coefficients of most of the residuals derived from the first-stage land investment equations are significantly different from zero. Moreover, the Wald chi-square test revealed that the residuals are jointly and significantly different from zero. Thus, the null hypothesis that the investment variables are exogenous in the exposure equations was rejected. These suggested simultaneity between direct and indirect exposure to LSLA and investment in land-improving techniques. This also implied that the coefficients of the investment variables would have been different if we had not controlled for such endogeneity. Thus, the inclusion of the residuals of land-improving techniques corrected for the endogeneity. Additionally, the $\chi 2$ statistics for significance of the instrument failed to reject the exclusion restriction that households' prior knowledge of other households affected by LSLA can only influence direct and indirect exposure to LSLA through farm investment.

Of particular interest were the results displayed by variables representing investment in land improving techniques. Most of these variables were negatively signed, as expected, and were significant at 1% in the equations for direct and indirect exposure to LSLA by domestic and foreign entities. Such findings supported past research (e.g., [64]) which found that farm investments enhanced tenure security of land and were less likely to experience exposure to LSLA. The results confirmed the reverse causality between exposure to direct and indirect exposure to LSLA and farm investment in northern Ghana. This finding further lends support to empirical and theoretical studies which have argued that farmers in many traditional tenure systems may minimize eviction rates through investment in land-improving techniques (e.g., [49,55,57]).

Further, gender, leadership position, and educational level were each appropriately signed and significant, and thus suggest that the probability of direct and indirect exposure to LSLA by domestic and foreign entities tends to be lower among male-headed households, leaders, and highly educated individuals as compared to non-exposure. This further suggests the importance of intrahousehold power dynamics in curtailing households' exposure to LSLA.

**Table 3.** Multinomial logit estimates of determinants of exposure to LSLA.

| Variable | Indirect Exposure to LSLA by FE | | Direct Exposure to LSLA by FE | | Indirect Exposure to LSLA by DE | | Direct Exposure to LSLA by DE | |
|---|---|---|---|---|---|---|---|---|
| | Coef. | SE | Coef. | SE | Coef. | SE | Coef. | SE |
| Well | −2.40 | 1.31 | −1.91 | 1.33 | −1.43 | 1.34 | −2.48 * | 1.30 |
| Local_catchments | −0.20 | 0.12 * | −0.27 | 0.13 ** | −0.29 | 0.04 *** | −0.10 | 0.05 ** |
| Drip_irrigation_facilities | −0.51 | 0.20 *** | 0.68 | 0.08 *** | −0.39 | 0.03 *** | 0.78 | 0.37 ** |
| Intercropping | −0.48 | 0.16 *** | 0.35 | 0.07 *** | −0.22 | 0.03 *** | −0.15 | 0.04 *** |
| Minimum tillage | −0.17 | 0.02 *** | −0.82 | 0.14 *** | −0.24 | 0.01 *** | −0.66 | 0.08 *** |
| Residue_retention | −0.02 | 0.01 ** | −0.04 | 0.02 ** | −0.03 | 0.01 *** | −0.24 | 0.09 ** |
| NPK | −0.71 | 0.17 *** | −0.75 | 0.21 *** | −0.29 | 0.15 * | 7.33 | 0.96 *** |
| SOA | −2.25 ** | 1.07 | −2.25 ** | 1.08 | −2.29 ** | 1.08 | −1.79 * | 1.08 |
| Urea | −1.50 | 1.01 | 1.93 * | 1.02 | 1.951 * | 1.04 | 1.02 | 1.60 |
| gr2_wells | 0.39 | 0.22 * | 0.71 | 0.26 ** | 0.57 | 0.20 *** | −0.40 | 0.14 *** |
| gr2_local_catchments | −0.18 | 0.10 * | −0.15 | 0.05 *** | 0.57 | 0.20 *** | 1.85 | 1.15 |
| gr2_drip_irrigation_facilities | −0.68 | 0.20 *** | 0.72 | 0.16 *** | −0.40 | 0.14 *** | 0.27 | 0.43 |
| gr2_intercropping | −0.42 | 0.12 *** | −0.17 | 0.39 | −0.14 | 0.09 | −0.12 | 0.38 |
| gr2_minimum_tillage | −0.40 | 0.16 ** | −1.62 | 1.66 | −0.01 | 0.01 | 0.20 | 0.02 *** |
| gr2_residue_retention | −0.26 | 0.16 * | −1.43 | 0.18 *** | 1.17 | 0.39 *** | −0.01 | 0.21 |
| gr2_npk | −0.38 | 0.19 ** | −0.71 | 0.17 *** | −0.75 | 0.21 *** | −0.29 | 0.15 * |
| gr2_soa | 0.01 | 0.01 | 1.27 | 0.60 ** | 3.77 | 1.29 *** | 2.86 | 1.23 ** |
| gr2_urea | −0.25 | 0.09 *** | 6.63 | 0.92 *** | −0.01 | 0.00 *** | 0.56 | 0.38 |
| HH_income | −0.08 | 0.13 | 0.21 | 0.21 | −0.01 | 0.01 | 0.01 | 0.12 |
| Leadership position | −0.04 | 0.01 *** | −0.20 | 0.12 * | −0.04 | 0.02 ** | −0.11 | 0.06 ** |
| Gender | −0.01 | 0.01 * | −0.11 | 0.03 *** | −0.41 | 0.17 ** | −0.02 | 0.12 |
| Age | −0.26 | 0.17 | −0.04 | 0.27 | 0.21 | 0.17 | 0.08 | 0.12 |
| Household size | 0.12 | 0.44 | 0.17 | 0.77 | −0.33 | 0.41 | −0.13 | 0.12 |
| Education | −0.05 | 0.00 *** | −0.09 | 0.00 *** | −0.35 | −0.22 | −0.12 | −0.16 |

**Table 3.** *Cont.*

| Variable | Indirect Exposure to LSLA by FE | | Direct Exposure to LSLA by FE | | Indirect Exposure to LSLA by DE | | Direct Exposure to LSLA by DE | |
|---|---|---|---|---|---|---|---|---|
| Farm size | −0.03 | 0.12 | 0.03 | 0.19 | −0.28 | 0.35 | 0.19 | 0.24 |
| Land institution | −0.01 | 0.01 * | −0.07 | 0.00 *** | −0.12 | 0.07 * | −0.28 | 0.12 ** |
| Social group | −0.15 | 0.09 * | −0.55 | 0.17 *** | −1.18 | 0.69 * | −0.28 | 0.15 * |
| Road | 0.18 | 0.30 | 0.48 | 0.55 | −0.45 | 0.57 | 0.44 | 0.39 |
| Credit | −0.11 | 0.74 | −0.35 | 0.50 | −0.50 | 0.11 *** | −0.17 | 0.04 *** |
| Water source | 0.29 | 0.02 *** | 0.21 | 0.03 *** | 0.11 | 0.23 | 0.43 | 0.55 |
| Good fertile [1] | 0.52 | 0.04 *** | 0.16 | 0.00 *** | 0.38 | 0.05 *** | 0.64 | 0.33 * |
| Moderate fertile [1] | 1.20 | 0.04 *** | 0.38 | 0.14 ** | −0.76 | 0.33 ** | 0.47 | 0.28 * |
| _cons | −1.04 | 0.08 *** | −0.40 | 0.08 *** | −1.99 | 1.07 * | −0.56 | 0.13 *** |
| Joint significance of location variables: $\chi^2$ (6) | 65.55 *** | | | | | | | |
| $\chi^2$ (9)- joint significance test of residuals | 6.45 *** | | | | | | | |
| Test of significance of instrument | 0.28 (0.99) | | | | | | | |
| No. of observations | 636 | | | | | | | |

Notes: ***, **, and * indicate statistical significance at 1, 5, and 10%. [1] Poorly fertile is reference category in all estimations; FE and DE represent foreign and domestic entities. Source: Author's Computation from Field Survey, 2018.

Land institutions also had the expected negative sign, and significantly influenced both direct and indirect exposure under domestic and foreign entities; thus, suggesting that availability of formal land institutions, such as Lands Commission and Land Use and Spatial Planning Department, are less likely to experience either direct or indirect exposure to LSLA under domestic and foreign entities. This is plausible because the information from the land institution can help facilitate the rate of registration, certification, security of land tenure rights, and, by extension, reduce the rate of eviction. This result is consistent with the finding of [50] in which show that land governance is an important determinant of LSLA in developing economies.

Social group membership was also plausibly signed and had a significant influence on direct and indirect exposure to LSLA under domestic and foreign entities, and thus, suggested that households that participate in social group activities are less likely to face direct or indirect exposure to LSLA. Ideally, the resurgence of LSLA after 2008 has seen Non-Governmental Organizations (NGOs) and other social groups, including Action Aid-Ghana, Regional Advocacy, and Information Network Systems (RAINS), at the forefront of the fight against acquirers. These groups express concerns on the food security implications of LSLA, and therefore work on slogans such as 'environmental protection watchdogs' and 'guardians of livelihoods of the poor' (e.g., [28]). The ambassadors for the marginalized have subscribed to some of these discourses in their campaign against LSLA. It is therefore not surprising that membership to a social group is significant and negatively related to direct and indirect exposure to LSLA under domestic and foreign entities.

Access to credit was also negatively signed in all equations as expected, but was significantly related to only direct and indirect exposure to LSLA by domestic entities. This suggests that households with credit access are less likely to lose land, directly or indirectly, to LSLA by domestic entities. The main reason ascribed to the insignificant relationship between credit access and exposure to LSLA under foreign entities is that credit received is not resourceful enough to contest acquisitions by wealthy foreigners who have the backing of state and traditional authorities.

On the other hand, the availability of water resources in the village where the household is located was also positively signed as expected, and yet was significantly related to only direct and indirect exposure to LSLA by foreign entities. This suggests that households located in villages with available water resources are more likely to experience direct and indirect exposure to LSLA by foreign entities, but not by domestic entities. One reason ascribed to this result is that domestic entities mainly rely on rainwater for production after acquisition, and therefore do not depend on other water resources before acquiring land for agricultural production. Soil fertility was also positively signed as expected, and was significantly related to direct and indirect exposure to LSLA under domestic and foreign entities. This suggests that households on plots with fertile soil are more likely to experience direct and indirect exposure to LSLA by domestic and foreign entities. This finding contrasts with those who argued that LSLA occurs only on marginal lands [28]. Lastly, the results also indicated that the location of the household also matters in its exposure to LSLA in the area. This is indicated by the joint significant test of the district dummies in Table 2. This probably shows the importance of agro-climatic variation and infrastructural differences in attracting investors of LSLA, and thus, confirms previous studies [50] highlighting the role of location in LSLA.

### 6.2. Farm Investments

As mentioned in Section 5.2, investment decisions in land- and yield-improving techniques may be interdependent (substitutes or complements). Thus, under the control function approach, the direct and indirect exposure variables and their corresponding residuals from a set of first stage probit models were incorporated into the second stage multivariate probit model. The first stage results of the determinants of LSLA are presented in Table A2 of the Appendix A, but are not discussed for the purposes of brevity. Furthermore, details of the determinants of exposure to LSLA are already presented and

discussed in Section 6.1. However, the Chi-square ($\chi^2$) statistics for significance of the instrument failed to reject the exclusion restriction that availability of formal land institutions, such as the Lands Commission and the Land Use and Spatial Planning Department, can only influence investment through direct and indirect exposure to LSLA by domestic and foreign entities.

Regarding the second stage multivariate probit model, we found that the likelihood ration test [$\chi^2(36) = 302.435$, $p = 0.000$] rejected the null hypothesis of zero correlation between the error terms, and thus, suggested that it is more efficient to jointly rather than separately model investment decisions. It is also worth mentioning that the results were mixed in terms of the signs of correlation coefficients between investment decisions in the land-improving techniques (Table 4). For instance, a negative correlation existed between investment in wells, local catchments, and drip irrigation facilities. However, each of these techniques was positively correlated with intercropping, probably because intercropping is relatively less expensive. The negative correlation between some of the decisions indicated that farmers perceive tradeoffs or consider these technologies as substitutes. This is expected, because some of the techniques serve the same purpose but are expensive in combination. Thus, farmers may choose between the three to minimize costs. On the other hand, the positive correlation indicated complementarity between the investment decisions. Nonetheless, the results were consistent with past studies (e.g., [62]) which showed interdependencies between households investment decisions on land- and yield-improving techniques.

**Table 4.** Correlation matrix from the multivariate probit model.

| | $\rho_W$ | $\rho_{LC}$ | $\rho_{DIF}$ | $\rho_I$ | $\rho_{MT}$ | $\rho_{RR}$ | $\rho_{NPK}$ | $\rho_{SOA}$ | $\rho_U$ |
|---|---|---|---|---|---|---|---|---|---|
| $\rho_W$ | 1 | | | | | | | | |
| $\rho_{LC}$ | −0.436 (0.07) *** | 1 | | | | | | | |
| $\rho_{DIF}$ | −0.177 (0.062) *** | −0.351 (0.077) *** | 1 | | | | | | |
| $\rho_I$ | 0.016 (0.063) | 0.333 (0.071) *** | 0.126 (0.065) * | 1 | | | | | |
| $\rho_{MT}$ | 0.018 (0.072) | 0.482 (0.075) *** | −0.087 (0.075) | −0.198 (0.073) *** | 1 | | | | |
| $\rho_{RR}$ | −0.082 (0.071) | 0.135 (0.084) | 0.087 (0.072) | −0.051 (0.072) | 0.063 (0.082) | 1 | | | |
| $\rho_{NPK}$ | 0.499 (0.052) *** | 0.246 (0.075) *** | −0.124 (0.067) * | −0.012 (0.066) | 0.189 (0.071) *** | 0.279 (0.069) *** | 1 | | |
| $\rho_{SOA}$ | 0.101 (0.066) | 0.002 (0.086) | 0.062 (0.069) | 0.092 (0.069) | 0.094 (0.073) | 0.119 (0.074) | 0.014 (0.068) | 1 | |
| $\rho_U$ | 0.253 (0.059) *** | 0.023 (0.079) | −0.160 (0.064) ** | 0.331 (0.060) *** | 0.048 (0.071) | −0.245 (0.068) *** | 0.218 (0.063) *** | −0.062 (0.067) | 1 |

Likelihood ratio test of

$\rho_{LCW} = \rho_{DIFW} = \rho_{IW} = \rho_{MTW} = \rho_{RRW} = \rho_{NPKW} = \rho_{SOAW} = \rho_{UW} = \rho_{DIFLC} = \rho_{ILC} = \rho_{MTLC} = \rho_{RRLC} = \rho_{NPKLC} = \rho_{SOALC} = \rho_{ULC} = \rho_{IDIF}$
$= \rho_{MTDIF} = \rho_{RRDIF} = \rho_{NPKDIF} = \rho_{SOADIF} = \rho_{UIF} = \rho_{MTI} = \rho_{RRI} = \rho_{NPKI} = \rho_{SOAI} = \rho_{UI} = \rho_{RRMT} = \rho_{NPKMT} = \rho_{SOAMT} = \rho_{UMT} = \rho_{NKPRR} = \rho_{SOARR}$
$= \rho_{URR} = \rho_{SOANPK} = \rho_{UNPK} = \rho_{USOA} = 0$

$X^2(36) = 302.435$, Prob > chi2 = 0.0000

*, **, and *** indicate statistical significance at 10, 5, and 1%, respectively; numbers in parentheses are the standard errors.

The results of the farm investment impact of LSLA by domestic and foreign entities are presented in Table 5. It is worth mentioning that the residuals derived from first-stage regressions for exposure to LSLA are statistically significant at conventional levels. These indicate a simultaneity bias. Shown in Table 5 are the Chi-square ($\chi^2$) statistics for the joint Wald tests on the vector of these residuals from the first-stage estimations. These values

reveal that for each investment equation, the null hypothesis that the residuals are jointly equal to zero is rejected. These confirm the results of the individual t-statistics, which indicate a simultaneity bias.

The results also revealed that most of the coefficients of direct and indirect exposure to LSLA by domestic and foreign entities were significant at 1%. However, these variables were negatively related to investment in the construction of wells, local catchments, drip irrigation facilities, minimum tillage, and residue retention, but positively related to investment NPK and Urea. This implied that households exposed to LSLA by domestic and foreign entities were less likely to invest in the construction of wells, local catchments, drip irrigation facilities, minimum tillage, and residue retention, but were more likely to invest in NPK and Urea. This is plausible given that investment in wells, local catchments, drip irrigation facilities, minimum tillage, and residue retention are not only relatively expensive, but long-term investments with longer time dimensions—at least one year—required to reap the benefits of investment. A follow-up during focus group discussions revealed that NKP and Urea have been subsidized in the area, and hence, the increase in investment. With the exception of results for fertilizers, such as NPK and Urea, the rest of the results were inconsistent with [14], who found positive spillovers on adoption of agricultural practices. On the other hand, direct and indirect exposure to LSLA by domestic and foreign entities were negatively related to investment in SOA, probably because of the negative effects on the soil. Ideally, ammonia disposition results in nitrogen leaching, eutrophication of groundwater, and soil acidification. Thus, investment in such techniques is likely to be lower among households even in times of shocks. Nonetheless, our findings generally support the notion that direct and indirect exposure to LSLA will compel farmers to choose short-term investments over long-term farm investments [5,45].

In addition to the above, household income was positively signed and significantly influenced investment in all land-improving techniques, except for investment in Sulphate of Ammonia (SOA), where a negative sign was observed. Thus, high-income households stand the chance of enhancing investment in wells, local catchments, drip irrigation facilities, intercropping with nitrogen-fixing crops, minimum tillage, residue retention, NKP, and Urea. This is not surprising, since rich households can still invest in both land and land-improving techniques in the wake of shocks. However, such rich households are less likely to invest in SOA. A group interview revealed that SOA leads to leaching, and moreover, is not covered by the fertilizer subsidy program under the Ghanaian governments' planting for food and jobs, which promotes the use of fertilizer among farmers. This result was consistent with past studies which highlighted the importance of wealth in farm investment [67].

On the other hand, households' participation in the fertilizer subsidy program was positive and significantly related to NPK and Urea. This suggested that participation in fertilizer subsidy programs will enhance investment in inorganic fertilizers, such as NPK and Urea. This was consistent with the finding of [68], which showed that participation in fertilizer subsidy programs is associated with an increase in the probabilities of inorganic fertilizer use in Tanzania. The main reason ascribed for the insignificant effect of participation in wells, local catchments, drip irrigation facilities, intercropping with nitrogen-fixing crops, minimum tillage, and residue retention is that the package of the fertilizer subsidy program does not cover the cost of irrigation or other soil and water techniques, and farmers lack the required resources to carry out such investments.

Table 5. Multivariate probit estimates of determinants of farm investments.

| VARIABLES | W | LC | DIF | I | MT | RR | NPK | SOA | Urea |
|---|---|---|---|---|---|---|---|---|---|
| Indirect_Foreign | −0.434 *** | −1.165 *** | −1.100 * | −0.085 | −0.276 * | −0.496 *** | 1.192 *** | −0.017 | 1.057 * |
| | (0.132) | (0.317) | (0.660) | (0.742) | (0.140) | (0.190) | (0.442) | (0.780) | (0.600) |
| Direct_Foreign | −0.363 ** | −0.624 *** | −0.617 ** | −0.281 | −0.746 * | −0.506 * | 0.448 * | −0.733 * | 1.073 ** |
| | (0.148) | (0.210) | (0.263) | (0.271) | (0.398) | (0.297) | (0.239) | (0.412) | (0.435) |
| Indirect_Domestic | −1.280 ** | −1.585 *** | −0.625 * | −0.259 | −0.370 *** | −0.191 *** | 0.391 ** | −0.557 ** | 2.039 ** |
| | (0.582) | (0.465) | (0.367) | (0.272) | (0.146) | (0.020) | (0.178) | (0.273) | (1.023) |
| Direct_Domestic | −0.589 *** | −0.679 *** | −0.923 *** | −0.474 | −1.430 *** | −1.586 * | 2.722 *** | −1.262 *** | 1.344 * |
| | (0.191) | (0.158) | (0.212) | (0.801) | (0.417) | (0.886) | (0.829) | (0.243) | (0.800) |
| gr2_DEDE | −0.900 * | 0.558 | 0.183 * | 0.635 ** | 0.624 | 0.822 | 1.363 *** | 0.051 | −0.114 |
| | (0.469) | (0.581) | (0.094) | (0.291) | (0.557) | (0.536) | (0.496) | (0.512) | (0.477) |
| gr2_IEDE | 0.915 ** | 0.906 ** | −0.286 | 0.580 *** | −0.616 *** | −0.604 *** | −0.589 *** | 0.575 | 1.285 ** |
| | (0.376) | (0.382) | (0.595) | (0. 226) | (0.059) | (0.131) | (0. 137) | (0.379) | (0.625) |
| gr2_IEFE | −0.0554 | −0.405 *** | −0.721 *** | 0.428 *** | 0. 476 *** | 0. 459 | 0.435 | 0. 460 *** | 0. 426 *** |
| | (0.417) | (0.129) | (0.247) | (0.073) | (0.147) | (0.354) | (0. 582) | (0.082) | (0.106) |
| gr2_DEFE | 0.491 *** | −0.334 | 0.897 *** | −0.018 | −0.673 | −0.487 ** | −0.446 *** | −0.394 | −0.412 *** |
| | (0.145) | (0.589) | (0.259) | (0.455) | (0.560) | (0. 220) | (0.031) | (0.468) | (0.143) |
| Household_income | 0.115 *** | 0.147 *** | 0.119 * | 0.249 ** | 0.134 * | 0.263 ** | 0.117 *** | −0.121 *** | 0.154 * |
| | (0.011) | (0.019) | (0.071) | (0.117) | (0. 073) | (0.128) | (0. 012) | (0.019) | (0.081) |
| Fert_subsidy | 0.0142 | 0.297 | −0.091 | 0.129 | 0.165 | 0.005 | 0.443 *** | −0.025 | 0.142 * |
| | (0.142) | (0.182) | (0.145) | (0.146) | (0.146) | (0.160) | (0.164) | (0.149) | (0.081) |
| Gender | 0.270 ** | 0.236 ** | 0.511 *** | 0.332 *** | −0.379 *** | −0.339 ** | 0.543 *** | 0.076 | 0.259 * |
| | (0.119) | (0.113) | (0.136) | (0.089) | (0.074) | (0.150) | (0.133) | (0.351) | (0.091) |
| Age_HHH | 0.003 | 0.000 | 0.000 | −0.005 | 0.012 ** | 0.005 | −0.003 | −0.002 | 0.011 |
| | (0.004) | (0.005) | (0.004) | (0.004) | (0.005) | (0.005) | (0.004) | (0.004) | (0.015) |
| HHsize | −0.000 | 0.001 | 0.006 | 0.013 | −0.009 | −0.013 | 0.005 | −0.007 | 0.000 |
| | (0.008) | (0.011) | (0.008) | (0.008) | (0.009) | (0.009) | (0.008) | (0.009) | (0.008) |
| Education | −0.006 | 0.104 *** | −0.016 | 0.007 | 0.003 | 0.033 ** | 0.011 ** | −0.023 | 0.190 ** |
| | (0.015) | (0.019) | (0.015) | (0.016) | (0.018) | (0.016) | (0.004) | (0.016) | (0.085) |
| Farm_size | 0.117 | 0.034 | 0.003 | 0.112 | 0.179 | 0.024 | 0.285 | −0.145 | 0.007 |
| | (0.149) | (0.156) | (0.153) | (0.151) | (0.172) | (0.168) | (0.352) | (0.158) | (0.150) |
| Compensation | 0.361 | 0.493 | 0.518 | −0.710 | −0.369 | −0.366 | 0.371 | −0.136 | −0.356 |
| | (0.400) | (0.565) | (0.423) | (0.463) | (0.424) | (0.430) | (0.403) | (0.411) | (0.399) |
| | 0.166 *** | 0.381 ** | 0.023 | 0.103 | 0.125 *** | 0.045 | 0.006 | 0.114 | 0.261 *** |
| | (0.042) | (0.189) | (0.126) | (0.124) | (0.044) | (0.139) | (0.125) | (0.129) | (0.062) |

**Table 5.** *Cont.*

| VARIABLES | W | LC | DIF | I | MT | RR | NPK | SOA | Urea |
|---|---|---|---|---|---|---|---|---|---|
| ROI | 0.148 | 0.029 | 0.114 ** | 0.256 | 0.169 | 0.014 | 0.259 * | 0.353 ** | 0.241 * |
| | (0.148) | (0.185) | (0.045) | (0.247) | (0.224) | (0.160) | (0.151) | (0.154) | (0.143) |
| Credit_access | −0.110 | −0.265 | −0.352 ** | −0.436 *** | −0.147 | −0.114 | −0.002 | −0.039 | −0.225 * |
| | (0.160) | (0.188) | (0.159) | (0.156) | (0.174) | (0.168) | (0.154) | (0.161) | (0.130) |
| Access_good_roads | 0.035 | −0.079 | 0.402 ** | −0.012 | 0.144 | 0.211 | 0.399 ** | 0.169 * | 0.302 * |
| | (0.156) | (0.161) | (0.197) | (0.159) | (0.178) | (0.173) | (0.157) | (0.100) | (0.156) |
| Tenure_security | 0.118 | 0.977 ** | 0.028 | 0.186 * | 0.215 ** | 0.251 ** | 0.115 * | 0.111 | 0.183 *** |
| | (0.234) | (0.472) | (0.191) | (0.101) | (0.086) | (0.104) | (0.063) | (0.194) | (0.028) |
| Good_fertile | −0.019 | −0.580 *** | −0.011 | −0.192 | −0.092 | −0.238 ** | −0.137 * | −0.185 | −0.135 *** |
| | (0.135) | (0.178) | (0.138) | (0.137) | (0.156) | (0.105) | (0.078) | (0.145) | (0.023) |
| Moderate_fertile | −0.089 | −0.312 | −0.219 | −0.240 | 0.115 | 0.034 | −0.090 | 0.197 | −0.019 |
| | (0.173) | (0.211) | (0.172) | (0.176) | (0.202) | (0.188) | (0.175) | (0.186) | (0.170) |
| Water_resource | 0.069 | 0.087 | −0.195 | −0.185 | 0.024 | −0.202 | −0.416 ** | 0.218 | 0.163 |
| | (0.170) | (0.219) | (0.173) | (0.173) | (0.201) | (0.186) | (0.173) | (0.182) | (0.174) |
| District_dummies | Yes | Yes | Yes | Yes | Yes | Yes | Yes | Yes | Yes |
| Constant | −0.359 | −1.804 ** | −1.233 * | 1.573 ** | −1.402 ** | −0.232 | −0.946 | −0.766 | −0.446 |
| | (0.635) | (0.875) | (0.650) | (0.670) | (0.702) | (0.670) | (0.644) | (0.654) | (0.624) |
| Joint significance of residuals: $\chi^2$ (4) | 25.23 *** | 31.45 *** | 28.27 *** | 42.23 *** | 19.80 *** | 27.11 *** | 34.56 *** | 21.33 *** | 22.45 *** |
| Joint significance of instruments: $\chi^2$ (1) | 1.497 [0.471] | 3.442 [0.121] | 1.691 [0.515] | 1.630 [0.110] | 0.982 [0.663] | 2.334 [0.348] | 2.22 [0.123] | 0.21 [0.100] | 1.89 [0.981] |
| Observations | 636 | 636 | 636 | 636 | 636 | 636 | 636 | 636 | 636 |

Notes: Standard errors in parentheses; *p*-values in squared brackets. *** $p < 0.01$, ** $p < 0.05$, * $p < 0.1$. W, LC, DIF, I, MT, RR, NPK, SOA, and Urea represent wells, local catchments, drip irrigation facilities, intercropping with nitrogen-fixing crops, minimum tillage, residue retention, NPK (15:15:15), Sulphate of Ammonia, and Urea (46:0:0), respectively; (Source: Author's Computation from Field Survey, 2018).

Gender of the head of the household had a positive and significant influence on wells, local catchments, drip irrigation facilities, intercropping with nitrogen-fixing crops, NKP, and Urea, and thus, indicated that male-headed households were more likely to invest in these land-improving techniques than female-headed households. This is plausible since males remain caretakers of most resources compared to women [56,69], and are therefore likely to invest resources in wells, local catchments, drip irrigation facilities, intercropping with nitrogen-fixing crops, minimum tillage, residue retention, NKP, Sulphate of Ammonia, and Urea, which are resource-intensive. This result was consistent with [56] who found a positive effect of gender on farm investment in Peru. On the other hand, gender was negative and significantly related to minimum tillage and residue retention. This suggested that investment in such techniques was more likely to be common among female-headed households compared to their male counterparts. A group discussion revealed that minimum tillage and residues retention are relatively cheaper to use and less expensive in the area compared to the other land-improving techniques.

Furthermore, level of education was positive and significantly related to investments in local catchment, residue retention, NPK, and Urea. These findings imply that highly educated households are more likely to enhance their probability of investment in these techniques. The results are plausible because education is assumed to increase households' ability to obtain, process, and use any accessible information related to farming and investment [67]. Highly educated households access more information about land-improving technologies, and therefore, stand the chance to invest in these practices than the less educated households. The results are consistent with past studies that examined the relationship between education and investment decisions in Africa [49].

Additionally, social group membership was positive and significantly related to construction of wells, local catchments, minimum tillage, and Urea. These findings implied households with a social network were more likely to enhance their probability of investment in these techniques compared to households without social networks. A similar result was found in a study by [67], in which group membership was found to have a positive impact on investment in organic soil amendments. Ideally, farmers' social networks tend to improve his/her access to production and investment information. Aside from information, such networks can lead to the formation of cooperative labor, where members constitute themselves into groups and take turns to provide the required labor for investing in land-improving technologies [69].

Return on investment (ROI) was significant and positively related to investment in drip irrigation facilities, NPK, SOA, and Urea, and thus, suggest that increasing returns from investment in such land-improving techniques is more likely to enhance household investment in such techniques. A focus group of participants explained that techniques such as drip irrigation facilities, NPK, SOA, and Urea are costly, and are therefore adopted so long as returns increase.

Similarly, access to road networks was also significant and positively related to investment in drip irrigation facilities, NPK, SOA, and Urea, and thus, suggest that households with road networks are more likely to invest in such techniques. This was consistent with past studies [67]. A focus group of participants explained that techniques such as drip irrigation facilities, NPK, SOA, and Urea depend on market access for procurement, and are therefore common among households with market access as indicated by access to road networks.

Credit enhances resource mobilization and investment in land-improving technologies [69]. Households with credit access were therefore expected to participate in investments in land-improving techniques. Surprisingly, access to credit was significant but did not have a positive sign. This suggests that households with access to credit were less likely to invest in land-improving technologies considered in this study. A follow-up focus group discussion revealed that most of the farmers do not have land, and therefore divert credit into other activities aside from the mobilization of resources for farming. This result

was consistent with [70], who investigated a significant relationship between credit and resource adoption in Africa.

Additionally, tenure security was found to have a positive and significant influence on investment in all land-improving techniques, and thus, suggests that land registration programs and policies can facilitate households' farm investment by enhancing tenure security. This was consistent with [5] in Thailand.

Furthermore, all the variables representing the fertility of soil had a negative sign, but most did not have a significant influence on investments. Thus, the fact that fertile plots do not enhance investment is explained to some degree. These results are similar to the study of [20] in Ghana.

Lastly, a joint significant test of the district dummies indicated that the districts jointly influence investment in land-improving techniques in the area. This shows the importance of location in the adoption of land-improving techniques, and thus, lends support to previous studies [70] that advocated for considerations of location in promoting land-improving technologies in Africa.

The empirical analysis revealed that investments in land-improving techniques by a given household significantly vary with direct and indirect exposure to LSLA under domestic and foreign entities. This, therefore, confirms the theoretical predictions that LSLA may increase the rate of evictions, enhance tenure insecurity, and compel farmers to prioritize investment in some land-improving techniques over other land-improving techniques (e.g., [45]). However, the results contradict other studies (e.g., [43,44]) which showed that LSLA may lead to increased investment in farm technologies by local communities. The results further revealed household income, participation in the fertilizer subsidy program, gender, education, and social group membership credit access as important determinants of long-term and short-term farm investment, and thus, suggest the importance of household and location characteristics in farm investment.

Our results also provided evidence that confirmed the notion that LSLA under domestic and foreign entities will decrease with investment in land-improving techniques. This lends support to the theoretical argument (e.g., [49,50])that there is a reverse causality between exposure to LSLA and farm investment where individual investments in land-improving techniques also contribute to lower tenure insecurity or rate of evictions.

In addition, household characteristics including gender, leadership position, educational level, social group membership, and credit access were revealed as important determinants of households' exposure to LSLA under domestic and foreign entities. Additionally, location characteristics such as the district of households, availability of land institution, availability of water resource, and fertility of soil tend to play a significant role in households' probability of being exposed to LSLA under domestic and foreign entities. These findings, therefore, revealed the strong role of household and location variables in LSLA.

### 6.3. Effect of LSLA on Level of Investments in Land-Improving Techniques

As previously stated, ordered probit regression with endogenous treatment was employed to estimate the effect of LSLA on level of investment in land-improving techniques. The model estimates the effects of exposure to LSLA on the level of investment, and the extra effect of exposure to LSLA in the presence of other factors influencing level of investments. In this section, our interest is the effects of direct and indirect exposure to LSLA by domestic and foreign entities on the level of investment, and as well as its effects through land tenure security (mediation effects). Thus, we present and discuss only the results of the effect of LSLA on level of investment, as well as its effects on investments through land tenure security (Tables 5–11).

**Table 6.** Correlation efficient test for endogeneity of exposure to LSLA.

| Parameter | Coefficient | Robust std. err. |
|---|---|---|
| corr(e.Direct_Foreign, e.oderedLIT) | 0.416 | 0.159 *** |
| corr(e.Indirect_Foreign, e.oderedLIT) | 0.505 | 0.203 ** |
| corr(e.Direct_Domestic,e.oderedLIT) | 0.855 | 0.162 *** |
| corr(e.Indirect_Domestic,e.oderedLIT) | 0.533 | 0.184 *** |

*** $p < 0.01$, ** $p < 0.05$.

**Table 7.** Average effect of exposure to LSLA on level of investment in land improving techniques (ATET).

| ATET-PrLI | Direct Exposure to LSLA by FE | Indirect Exposure to LSLA by FE | Direct Exposure to LSLA by DE | Indirect Exposure to LSLA by DE |
|---|---|---|---|---|
| ATET_Pr0 | 0.021 (0.022) | 0.152 (0.035) *** | 0.101 (0.102) | 0.056 (0.023) ** |
| ATET_Pr1 | −0.108 (0.035) *** | −0.235 (0.044) *** | −0.140 (0.038) *** | −0.390 (0.036) *** |
| ATET_Pr2 | −0.136 (0.055) ** | −0.047 (0.067) | −0.267 (0.046) *** | −0.198 (0.065) *** |
| ATET_Pr3 | −0.121 (0.049) ** | −0.032 (0.065) | −0.117 (0.039) *** | −0.197 (0.090) ** |
| ATET_Pr4 | −0.029 (0.041) | −0.098 (0.037) *** | −0.006 (0.040) | −0.006 (0.042) |
| ATET_Pr5 | −0.316 (0.113) *** | −0.304 (0.159) * | −0.631 (0.103) *** | −0.283 (0.149) * |

Notes: Robust standard errors in parentheses; *** $p < 0.01$, ** $p < 0.05$, * $p < 0.1$; ATET is average treatment effect on the treated; LI is level of investment, LSLA is large-scale land acquisition; FE and DE represent foreign entities and domestic entities, respectively; ATET_Pr0, ATET-Pr1, ATET-Pr2, ATET-Pr3, ATET-Pr4, and ATET-Pr5 represent average treatment effect (exposure effect) on the probability of noninvestment, investment in 1, 2, 3, 4, and 5 or more of any land improving techniques in this study.

**Table 8.** Effect of direct exposure to LSLA by foreign entities through tenure security.

| Variables | oderedLIT |
|---|---|
| 1.Perception_tenure_security#0b.Direct_Foreign | 0.577 *** |
| | (0.230) |
| 1.Perception_tenure_security#1.Direct_Foreign | −0.157 * |
| | (0.093) |
| Observations | 661 |

Notes: Robust standard errors in parentheses; *** $p < 0.01$, * $p < 0.1$; oderedLIT is level of investment in land improving techniques.

**Table 9.** Effect of indirect exposure to LSLA by foreign entities through tenure security.

| Variables | oderedLIT |
|---|---|
| 1.Perception_tenure_security#0b.Indirect_Foreign | 0.635 *** |
| | (0.113) |
| 1.Perception_tenure_security#1.Indirect_Foreign | −0.117 |
| | (0.274) |
| Observations | 661 |

Notes: Robust standard errors in parentheses; *** $p < 0.01$; oderedLIT is level of investment in land improving techniques.

**Table 10.** Effect of direct exposure to LSLA by domestic entities through tenure security.

| Variables | oderedLIT |
|---|---|
| 1.Perception_tenure_security#0b.Direct_Domestic | 0.843 *** |
| | (0.113) |
| 1.Perception_tenure_security#1.Direct_Domestic | −0.311 *** |
| | (0.111) |
| Observations | 661 |

Notes: Robust standard errors in parentheses; *** $p < 0.01$; oderedLIT is level of investment in land improving techniques.

**Table 11.** Effect of indirect exposure to LSLA by domestic entities through tenure security.

| Variables | oderedLIT |
| --- | --- |
| 1.Perception_tenure_security#0b.Indirect_Domestic | 0.839 *** |
| | (0.115) |
| 1.Perception_tenure_security#1.Indirect_Domestic | −0.414 * |
| | (0.250) |
| Observations | 661 |

Notes: Robust standard errors in parentheses. *** $p < 0.01$, * $p < 0.1$; oderedLIT is level of investment in land improving techniques.

The results of the extra effect of exposure to LSLA in the presence of other factors influencing the level of investments are presented in Tables S1–S4 of the Supplementary Materials. In Table 6, correlations between the errors from the level of investment in the land improving techniques and the errors from direct and indirect exposure to LSLA by domestic and foreign entities range between 0.42 and 0.86. These are all significantly different from zero, and are positively related. These, therefore, imply that direct and indirect exposure to LSLA by domestic and foreign entities are all endogenous in the equations for level of investment. These further imply that unobserved factors that increase the likelihood of households' direct and indirect exposure to LSLA by domestic and foreign entities also tend to increase the likelihood of investing in higher levels of land-improving techniques. Thus, the use of ordered probit regression with endogenous treatment is appropriate as linear or ordered probit regression would have produced biased results.

As shown in Table 7, we found that direct and indirect exposure to LSLA by domestic and foreign entities was likely to decrease the proportion of households investing at different levels of land-improving techniques. For instance, the treatment effect for direct and indirect exposure to LSLA by foreign and domestic entities (ATET_Pr1) ranged between 0.11 and 0.39 lower. This implied that the expected decrease in the probability of investing in any one of the land-improving techniques was 0.11–0.39 lower than it would be if households were not exposed to LSLA by domestic and foreign entities. Further, the treatment effect for direct and indirect exposure to LSLA by foreign and domestic entities (ATET_Pr2) ranged between 0.14 and 0.27 lower when compared to non-exposure to LSLA, and thus, indicates that the expected decrease in the probability of investing in any two of the land-improving techniques (ATET_Pr2) was 0.14–0.27 lower than it would be if households were not exposed to LSLA by domestic and foreign entities. Similarly, the average probability of investing in any three of the land-improving techniques (ATET_Pr3) was 0.12–0.20 lower than it would be if households were not directly or indirectly exposed to LSLA by domestic and foreign entities. For those households investing in any four of the land-improving techniques, only indirect exposure to LSLA by foreign entities was significantly related to households' level of investment. This implies that the average probability of investing in any four (ATET_Pr4) of the techniques was expected to be 0.10 lower than it would be if they were not directly exposed to LSLA by foreign entities. Lastly, the treatment effect for direct and indirect exposure to LSLA by foreign and domestic entities (ATET_Pr5) ranged between 0.28 and 0.63 lower compared to non-exposure to LSLA. Thus, the average probability of investing in any three of the land-improving techniques was 0.28–0.63 lower than it would be if households were not directly or indirectly exposed to LSLA by domestic and foreign entities. For those households that were not investing in any of the land improving techniques, only indirect exposure to LSLA by foreign and domestic entities was significant and positively related to households' level of investment. This implied that the average probability of not investing in any land-improving techniques was 0.15 and 0.06 higher for households that were indirectly exposed to LSLA by domestic and foreign entities. These results are not surprising, as land is the basis for agricultural-related activities, and losses due to acquisition will affect such activities, including the investment in land-improving techniques. Generally, these results contrast with the notion

that LSLA can bring about technology transmission and adoption among farm households (e.g., [7,43,44]).

Regarding the extra effect of direct and indirect exposure to LSLA by domestic and foreign entities in the presence of tenure security, estimates of the coefficients are shown in Tables 8–11. The results showed that the perception of tenure security had different effects for the exposed compared to the nonexposed. Table 8, for example, shows the effects of direct exposure to LSLA by foreign entities in the presence of tenure security. For non-exposure to LSLA by foreign entities, perception of tenure security (1.Perception_tenure_security#0b.Direct_Foreign) was significant and positively related to the level of investment in land-improving techniques. However, for direct exposure to LSLA by foreign entities, tenure security (1.Perception_tenure_security#1.Direct_Foreign) was negatively related to the level of investment in land-improving techniques. This implies that in the presence of tenure security, the level of investment in land-improving techniques was lower for households under direct exposure to LSLA by foreign entities, but higher for households that were not directly exposed to LSLA by foreign entities. Similarly, the level of investment in the presence of tenure security was lower for households under indirect exposure to LSLA by foreign entities, direct exposure to LSLA by domestic entities, and indirect exposure to LSLA by domestic entities. On the contrary, level of investment in the presence of tenure security was higher for households that were not directly or indirectly exposed to LSLA by domestic and foreign entities. Similar results are observed in Tables 9–11. These results, therefore, imply that in the absence of LSLA, an increase in perceptions of tenure security tends to enhance the level of investment in land-improving techniques. However, the emergence of LSLA will affect such perceptions, and this will in turn reduce the level of investment in land-improving techniques. Tenure insecurity brings about uncertainty in the farmer's mind as to whether he/she may be able to reap the full benefits of his/her investment. LSLA remains one of the central issues causing tenure insecurity because of the displacements associated with it [48]. Thus, tenure security and absence of LSLA among households was likely to enhance investment in land improving techniques, as observed in the results.

## 7. Conclusions

This study examined the relationship between LSLA and households' farm investment decisions and the level of investment in land-improving techniques in northern Ghana. Specifically, the study tested the relationships between direct and indirect exposure to LSLA under domestic and foreign entities and households' investment in land-improving techniques. A control function approach was employed on cross-sectional data of 664 households selected through a multi-stage sampling technique for the analysis. This analysis was born out of the neoclassical theory of the relationship between tenure security and farm investment, which argues that land scarcity increases tenure insecurity and compels farmers to prioritize investment in some land-improving techniques over other land-improving techniques. The results revealed that both direct and indirect exposure to LSLA under domestic and foreign entities dissipates investment in land-improving techniques, such as the construction of wells, local catchments, drip irrigation facilities, intercropping with nitrogen fixing crops, minimum tillage, and residue retention, but enhances investment in NPK and Ammonia. The results also revealed that investments in these techniques decreased the households' probability of direct and indirect exposure to LSLA under domestic and foreign entities. Thus, while LSLA dissipates farm investment, the reverse causality is also possible where farm investment reduces a household's likelihood of being exposed to LSLA by domestic and foreign entities. Based on the findings of the effect of direct and indirect exposure to LSLA from domestic and foreign entities, this study concludes that there is a bi-directional relationship between LSLA and households' farm investment. Thus, households that lose land, labor, land-based resources, and uncultivated land, or that live nearby affected households or have limited land due to enclosures by domestic and foreign entities, are more likely to choose short-term investments over long-term investments in northern

Ghana. Additionally, households that invest in long-term and short-term land-improving technologies are likely to avoid eviction by domestic and foreign entities in northern Ghana. Regarding the effect of LSLA on the level of investment in land-improving techniques, we found that direct and indirect exposure to LSLA by domestic and foreign entities decreases the probabilities of investment at all levels of land-improving techniques. Even in the presence of high perception of tenure security, LSLA was found to decrease households' level of investment in land-improving techniques, probably by first changing households' perceived tenure security and then discouraging investment at all levels of land-improving techniques. Thus, the increased perception of tenure security under customary or traditional systems of land governance is not reliable in the wake of LSLA by domestic and foreign entities. We therefore recommend land registration programs and policies than can reduce eviction rates and facilitate households' investment in land-improving techniques.

**Supplementary Materials:** The following supporting information can be downloaded at: https://www.mdpi.com/article/10.3390/land12040737/s1, Table S1: Determinants and extra effects of direct exposure to LSLA by foreign entities in the presence of other factors influencing levels of investments; Table S2: Determinants and extra effects of indirect exposure to LSLA by foreign entities in the presence of other factors influencing levels of investments; Table S3: Determinants and extra effects of direct exposure to LSLA by domestic entities in the presence of other factors influencing levels of investments; Table S4: Determinants and extra effects of indirect exposure to LSLA by domestic entities in the presence of other factors influencing levels of investments.

**Author Contributions:** Conceptualization, A.-H.A., M.A. and J.A.A.; methodology, A.-H.A.; software acquisition, A.-H.A.; validation, A.-H.A., M.A. and J.A.A.; formal analysis, A.-H.A.; investigation, A.-H.A.; data curation, A.-H.A.; writing- original draft preparation, A.-H.A.; writing- reviewing and editing, A.-H.A., M.A. and J.A.A.; visualization, A.-H.A.; supervision, M.A. and J.A.A.; project administration, A.-H.A., funding acquisition; A.-H.A., M.A. and J.A.A. All authors have read and agreed to the published version of the manuscript.

**Funding:** This research received no external funding.

**Institutional Review Board Statement:** Not applicable.

**Informed Consent Statement:** Informed consent was obtained from all subjects involved in the study.

**Data Availability Statement:** The data that support the findings of this study are available upon reasonable request from the corresponding author. The data are not publicly available because it contains information that could not only compromise research participant privacy but expose them to vilification and beratement for providing information to researchers investigating large-scale land acquisitions in northern Ghana.

**Conflicts of Interest:** The authors declare no conflict of interest.

## Appendix A

**Table A1.** First-stage estimations of determinants of household's investment in long and short-term land improving technologies.

| Variables | W | LC | DIF | I | MT | RR | NPK | SOA | Urea |
|---|---|---|---|---|---|---|---|---|---|
| Leadership_position | 0.128 * | 0.672 ** | 0.341 *** | 0.233 ** | 0.799 *** | −0.243 ** | 0.167 *** | −0.118 | 0.913 *** |
| | (0.071) | (0.333) | (0.105) | (0.104) | (0.120) | (0.117) | (0.059) | (0.108) | (0.103) |
| Gender | 0.00966 | −0.142 | 0.146 | 0.0117 | 0.117 | −0.181 | 0.0692 | 0.218 | −0.0924 |
| | (0.161) | (0.205) | (0.169) | (0.166) | (0.199) | (0.183) | (0.168) | (0.182) | (0.162) |
| Age_HHH | −0.355 *** | −0.102 * | −0.199 *** | 0.318 *** | 0.013 *** | 0.631 * | 0.505 ** | −0.153 *** | 0.009 ** |
| | (0.113) | (0.050) | (0.041) | (0.048) | (0.004) | (0.357) | (0.218) | (0.043) | (0.004) |

**Table A1.** *Cont.*

| Variables | W | LC | DIF | I | MT | RR | NPK | SOA | Urea |
|---|---|---|---|---|---|---|---|---|---|
| HHsize | −0.006 | −0.009 | 0.005 | −0.00322 | −0.004 | −0.012 | 0.005 | −0.009 | −0.002 |
| | (0.007) | (0.010) | (0.007) | (0.113) | (0.009) | (0.008) | (0.007) | (0.008) | (0.007) |
| Education | −0.000 | 0.022 | 0.019 | −0.005 | 0.007 | 0.029 ** | −0.008 | 0.014 | −0.002 |
| | (0.013) | (0.016) | (0.013) | (0.013) | (0.016) | (0.014) | (0.013) | (0.014) | (0.013) |
| Know_anybodylossland | 0.217 ** | 0.180 *** | 0.266 ** | 0.014 * | 0.908 *** | 0.310 ** | 0.146 * | 0.682 *** | 0.340 *** |
| | (0.106) | (0.045) | (0.109) | (0.007) | (0.131) | (0.130) | (0.087) | (0.119) | (0.112) |
| Social_group | 0.188 * | 0.237 ** | 0.212 * | 0.536 *** | 0.120 *** | 0.121 * | 0.223 * | −0.146 * | 0.557 *** |
| | (0.107) | (0.120) | (0.110) | (0.109) | (0.028) | (0.070) | (0.122) | (0.084) | (0.107) |
| Access_good_roads | −0.0408 | 0.166 | 0.172 *** | 0.159 ** | 0.088 | 0.137 | 0.873 *** | −0.166 *** | 0.525 *** |
| | (0.106) | (0.136) | (0.030) | (0.070) | (0.123) | (0.122) | (0.110) | (0.050) | (0.190) |
| Credit_access | 0.035 | −0.013 | −0.089 | 0.083 | −0.167 | 0.038 | 0.042 | 0.057 | 0.002 |
| | (0.111) | (0.140) | (0.114) | (0.108) | (0.126) | (0.123) | (0.110) | (0.113) | (0.107) |
| Water_resource | 0.303 *** | 0.160 ** | 0.142 *** | −0.149 | −0.123 | −0.131 | 0.106 *** | 0.216 | 0.202 * |
| | (0.109) | (0.074) | (0.030) | (0.132) | (0.154) | (0.146) | (0.033) | (0.141) | (0.120) |
| Good_fertile [1] | 0.066 | −0.346 ** | 0.009 | 0.220 ** | −0.711 *** | −0.202 | 0.635 *** | −0.151 | 0.538 *** |
| | (0.106) | (0.145) | (0.109) | (0.109) | (0.126) | (0.125) | (0.110) | (0.115) | (0.107) |
| Moderate_fertile [1] | −0.351 ** | −0.140 | −0.135 *** | −0.268 * | 0.006 | 0.006 | 0.015 | 0.178 | −0.025 |
| | (0.150) | (0.195) | (0.052) | (0.155) | (0.181) | (0.170) | (0.157) | (0.163) | (0.151) |
| Perception_tenure_security | 0.179 *** | 0.106 | 0.352 *** | 0.135 | 0.117 | 0.195 *** | 0.203 * | 0.050 | 0.187 *** |
| | (0.029) | (0.168) | (0.133) | (0.030) | (0.152) | (0.048) | (0.121) | (0.136) | (0.029) |
| Fert_subsidy | 0.068 | 0.208 | −0.018 | 0.041 | 0.270 ** | −0.200 | −0.036 | −0.062 | 0.054 |
| | (0.110) | (0.143) | (0.112) | (0.111) | (0.127) | (0.129) | (0.113) | (0.117) | (0.110) |
| Farm_size | 0.082 | 0.224 | −0.003 | 0.072 | 0.091 | −0.133 | 0.033 | −0.237 * | −0.089 |
| | (0.116) | (0.148) | (0.120) | (0.118) | (0.136) | (0.136) | (0.120) | (0.126) | (0.117) |
| District_dummies | Yes | Yes | Yes | Yes | Yes | Yes | Yes | Yes | Yes |
| Constant | −0.666 * | −1.246 * | −0.756 * | 1.066 ** | −1.389 ** | −0.541 | −0.547 | −0.607 | −0.138 |
| | (0.400) | (0.657) | (0.398) | (0.513) | (0.551) | (0.508) | (0.486) | (0.509) | (0.463) |
| Observations | 659 | 655 | 661 | 661 | 651 | 650 | 657 | 661 | 657 |

**Notes**: Standard errors in parentheses; *** $p < 0.01$, ** $p < 0.05$, * $p < 0.1$; W, LC, DIF, I, MT, RR, NPK, SOA and Urea represent wells, local catchments, drip irrigation facilities, intercropping with nitrogen fixing crops, minimum tillage, residue retention, NPK (15:15:15), Sulphate of Ammonia and Urea (46:0:0), respectively; [1] Poorly fertile is reference category in all estimations. Source: Author's Computation from Field Survey, 2018.

**Table A2.** First-stage estimations of determinants of direct and indirect exposure to land grabbing by domestic and foreign entities.

| Variables | IEDE | DEDE | IEFE | DEFE |
|---|---|---|---|---|
| Leadership_position | −0.454 *** | −0.868 *** | −0.815 *** | −0.186 *** |
| | (0.124) | (0.123) | (0.127) | (0.025) |
| Gender | −0.190 *** | −0.284 *** | −1.089 *** | 0.831 *** |
| | (0.014) | (0.011) | (0.184) | (0.247) |
| Age_HHH | −0.242 *** | −0.149 *** | −0.181 *** | 0.177 *** |
| | (0.004) | (0.004) | (0.009) | (0.002) |

**Table A2.** *Cont.*

| Variables | IEDE | DEDE | IEFE | DEFE |
|---|---|---|---|---|
| HHsize | 0.008 | −0.006 | −0.006 | −0.001 |
| | (0.009) | (0.008) | (0.009) | (0.009) |
| Education | v0.210 *** | −0.167 *** | 0.018 | −0.000 |
| | (0.017) | (0.016) | (0.016) | (0.016) |
| Access_formal_landinstite | −1.404 *** | −1.111 *** | −1.206 *** | −1.312 *** |
| | (0.253) | (0.193) | (0.216) | (0.225) |
| Social_group | 0.696 *** | −0.130 ** | −0.597 *** | −0.205 * |
| | (0.130) | (0.060) | (0.133) | (0.112) |
| Access_good_roads | 0.157 | 0.243 * | −0.371 *** | −0.165 |
| | (0.128) | (0.126) | (0.138) | (0.132) |
| Perception_tenure_security | −0.311 ** | −0.372 ** | −0.240 * | −0.495 *** |
| | (0.149) | (0.158) | (0.138) | (0.147) |
| Fert_subsidy | −0.065 | 0.294 ** | 0.0221 | −0.355 ** |
| | (0.135) | (0.129) | (0.135) | (0.141) |
| Good_fertile [1] | 0.131 * | 0.187 *** | 0.353 *** | 0.132 *** |
| | (0.04) | (0.028) | (0.129) | (0.016) |
| Moderate_fertile [1] | 0.115 | 0.012 | 0.413** | 0.144 |
| | (0.179) | (0.179) | (0.202) | (0.179) |
| Farm_size | 0.017 | 0.008 | −0.143 | 0.082 |
| | (0.141) | (0.157) | (0.144) | (0.153) |
| Credit_access | −0.278 ** | −0.328 ** | −0.059 | −0.003 |
| | (0.128) | (0.129) | (0.128) | (0.128) |
| Water_resource | 0.247 | 0.187 | 0.306 ** | 0.217 * |
| | (0.161) | (0.164) | (0.149) | (0.128) |
| Constant | −1.844 *** | −1.005 * | 0.566 ** | −1.258 * |
| | (0.705) | (0.563) | (0.226) | (0.712) |
| District_dummies | Yes | Yes | Yes | Yes |
| Observations | 661 | 661 | 661 | 661 |

**Notes**: Standard errors in parentheses; *** $p < 0.01$, ** $p < 0.05$, * $p < 0.1$; [1] Poorly fertile is reference category in all estimations; DE and IE are direct and indirect exposure to LSLA, respectively.

**Table A3.** Scale of arable land under commercial deals by district.

| District | Total Area under LSLA (ha) | % of Total Deals |
|---|---|---|
| Central Gonja | 30,989.92 | 43.17 |
| Mampurugu-Muagdure | 10,905.43 | 15.19 |
| Mion | 10,783.30 | 15.02 |
| Savelegu | 10,369.17 | 14.44 |
| Sagnarigu | 5,479.11 | 7.63 |
| North Gonja | 2452.26 | 3.42 |
| Bole | 466.82 | 0.65 |
| Tamale Metro | 173.24 | 0.24 |
| Gushiegu | 34.13 | 0.05 |
| Bunkpurugu-Yunyoo | 24.38 | 0.03 |
| Yendi Municipal | 23.1 | 0.03 |
| East Gonja | 20.32 | 0.03 |
| Nanumba South | 13.47 | 0.02 |
| Nanumba North | 13.36 | 0.02 |
| West Mampurisi | 12.59 | 0.02 |
| Saboba | 12.52 | 0.02 |
| Kpandai | 12.29 | 0.02 |
| Total | 71,785.41 | 100.00 |

**Source**: Authors compilation using data obtained from Regional Lands Commission, 2017.

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
