# Peer review of "Large-Scale Land Acquisition and Household Farm Investment in Northern Ghana"

_land, doi:10.3390/land12040737_

Round 1
Reviewer 1 Report
It was interesting to read this manuscript. The topic is definitely necessary and approach in this manuscript relevant. Lot of material has been worked through to compose the manuscript.
The structure of the paper needs modifications to make it more reader friendly.
Figure 1 is first mentioned in row 379 but the figure itself is presented in row 451. From the readers point of view it would be better if the figure comes right after it is first mentioned. I suggest to present the figure after the sentence that begins in row 378 and ends in row 389 (The first stage consisted of the selection of six districts (Figure 1) based on the predominance of vast tracks of arable land under commercial deals.)
Paragraphs should be shortened (divided into more shorter paragraphs). It is really difficult to read such long paragraphs of tight text.
One detailed figure about data collections could give better overview on how the data was collected, what kind of data was collected, how many households from which district was involved etc. I suggest to compose such figure. One example on how the figure simplifies understanding the use methods can be find here: https://www.mdpi.com/2077-0472/12/6/850/htm Figure 2.
Table 1 is first mentioned in row 419 but the table itself is in row 451. It would be better if the table is situated right after it is fist mentioned. Same with table 2.
Row 807: The results of the farm investment impact of direct and indirect exposure to LSLA by domestic and foreign entities are presented in Table 4. The table itself is in row 970. Before the table 4 Table 3 comes but is first mentioned after table 4. This needs to be fixed. All the figures and tables should come right after they are first mentioned in the text. Right now the reading and understanding the work is really difficult.
Author Response
Response to Reviewer 1 Comments and Suggestions on land-1967330
Reviewer Comments and Suggestions: It was interesting to read this manuscript. The topic is definitely necessary and approach in this manuscript relevant. Lot of material has been worked through to compose the manuscript.
The structure of the paper needs modifications to make it more reader friendly.
Response: Dear reviewer, thank you for making time from your busy schedules to review our manuscript. Please, the structure of the paper has been modified by appropriately labeling and placing Figure 1 and all the tables in the text. We have also added Figure 2 to provide more information about the location of the households interviewed. Thank you so much.
Reviewer Comments and Suggestions: Figure 1 is first mentioned in row 379 but the figure itself is presented in row 451. From the readers point of view it would be better if the figure comes right after it is first mentioned. I suggest to present the figure after the sentence that begins in row 378 and ends in row 389 (The first stage consisted of the selection of six districts (Figure 1) based on the predominance of vast tracks of arable land under commercial deals.)
Response: Thank you for this suggestion. Figure 1 is now appropriately placed following your suggestion that figure should come after the sentence that begins in row 378 and ends in row 389.
Reviewer Comments and Suggestions: Paragraphs should be shortened (divided into more shorter paragraphs). It is really difficult to read such long paragraphs of tight text.
Response: Thank you for this valuable comment. The lengthy paragraphs are now divided into short paragraphs.
Reviewer Comments and Suggestions: One detailed figure about data collections could give better overview on how the data was collected, what kind of data was collected, how many households from which district was involved etc. I suggest to compose such figure. One example on how the figure simplifies understanding the use methods can be find here: https://www.mdpi.com/2077-0472/12/6/850/htm Figure 2.
Response: Thank you for this suggestion. Figure 2 presented to provide more information about data collection and a better overview location of the households interviewed
Reviewer Comments and Suggestions: Table 1 is first mentioned in row 419 but the table itself is in row 451. It would be better if the table is situated right after it is fist mentioned. Same with table 2.
Response: Thank you for your suggestion. All the tables including Tables 1 and 2, are now appropriately labeled and placed in the text.
Reviewer Comments and Suggestions: Row 807: The farm investment impact of direct and indirect exposure to LSLA by domestic and foreign entities are presented in Table 4. The table itself is in row 970. Before the table 4 Table 3 comes but is first mentioned after table 4. This needs to be fixed. All the figures and tables should come right after they are first mentioned in the text. Right now, the reading and understanding the work is really difficult.
Response: Please, all the tables are now appropriately labeled and placed in the text. Thank you so much.
Thank you so much.

Reviewer 2 Report
This research focused on large-scale and acquisition and household farm investment in Northern Ghana. It is a solid work, which will provide insurance for investments in all land-improving techniques.
There are some details in the manuscript. Comments and Suggestions for Authors are as follows.
1. Line 58-65: there are too many literature in a sentence.
2. Line 332: Please delete “ 4.1.1 Hypotheses to be tested ”.
3. Line 465-467: “ Xi ” is not coordinate with other words in the sentence.
4. Line 671-677: the word “j” is not found in the Formula 12.
5. Line 1075-1076: Please delete the word “Conclusions”.
6. Figure 1, Table 1, Table 2 and Table 3 need to be further improved.
Author Response
Response to Reviewer 2 Comments and Suggestions on land-1967330
This research focused on large-scale and acquisition and household farm investment in Northern Ghana. It is a solid work, which will provide insurance for investments in all land-improving techniques.
There are some details in the manuscript. Comments and Suggestions for Authors are as follows.
Reviewer Comments and Suggestions 1: Line 58-65: there are too many literatures in a sentence.
Response: The literature in lines 58-65 has been reduced. Thank you.
Reviewer Comments and Suggestions 2: Line 332: Please delete “ 4.1.1 Hypotheses to be tested ”.
Response: Please section “4.1.1 Hypotheses to be tested” has been deleted for your kind attention and consideration.
Reviewer Comments and Suggestions 3: Line 465-467: “ Xi ” is not coordinate with other words in the sentence.
Response: We appreciate this correction. It was actually an oversight on our part and has now been rectified. Throughout the manuscript, “ Xi ” is not properly defined and linked with its coefficients in each model.
Reviewer Comments and Suggestions 4. Line 671-677: the word “j” is not found in the Formula 12.
Response: We appreciate this correction too. It has now been rectified. J has been replaced with the appropriate symbols in the manuscripts. In Formula 12, j has been replaced with the decision to invest in 5 or more of the techniques.
Reviewer Comments and Suggestions 5: Line 1075-1076: Please delete the word “Conclusions”.
Response: Please, this is not clear to us. Are we deleting the heading for “Conclusion” from the manuscript?
Reviewer Comments and Suggestions 6. Figure 1, Table 1, Table 2 and Table 3 need to be further improved.
Response: Thank you for this suggestion. All the Tables and Figures are appropriately labeled and placed in the text to improve clarity.
Thank you so much.

Reviewer 3 Report
1. There have been a lot of literature discussions on land property rights and investment, but the marginal contribution of this paper is not great. Write the Novelty of your study.
2. Although the author cited more than 100 papers, he neglected the related papers in recent important journals. He suggested that the author should supplement the literature reading and cite more high-level literature. Please write a historical perspective of your study in the literature review section. A critical analysis of the literature review is necessary.
3. Specify the theoretical background of the paper, especially the hypothesis.
4. Results of some sections are presented mechanically. Cross Connection is necessary (Is your finding similar to or contradicts other studies? If so why ?)
5. Please explain why “On the contrary, level of investment in the presence of tenure security is higher for households that are not directly or indirectly exposed to LSLA by domestic and foreign entities. These results, therefore, imply that in the absence of LSLA, an increase in perception of tenure security tends to enhance the level of investment in land-improving techniques. However, the emergence of LSLA will affect such perception and this will, in turn, reduce level of investment in land-improving techniques. ”
6.This manuscript requires proper proofreading and grammar checking.
Author Response
Response to Reviewer 3 Comments and Suggestions for Authors
Comments and Suggestions 1: There have been a lot of literature discussions on land property rights and investment, but the marginal contribution of this paper is not great. Write the Novelty of your study.
Response: Thank you so much for your comment. We are grateful. The novelty of this study comes in three folds and is stated as follows:
- First, the study examines the implications of LSLA on household farm investment with special reference to different actors involved in LSLA, and a particular interest in the exact impact they have on land- and yield-improving techniques.
- Second, the study extends the analysis to the implication of LSLA on the level of investment in land and yield-improving techniques since there are several investment levels and the possibility that LSLA does not affect them in the same way.
- This study extends the control function approach proposed by (Wooldridge, 2014, 2015) and the extended regression models (StataCorp, 2021), on detailed household-level survey data from the 2017/2018 cropping season, to account for potential endogeneity arising from both observed and unobserved heterogeneity and to assess the effects of LSLA on investment in interrelated techniques and as well as the level of investment in these techniques.
Comments and Suggestions 2: Although the author cited more than 100 papers, he neglected the related papers in recent important journals. He suggested that the author should supplement the literature reading and cite more high-level literature. Please write a historical perspective of your study in the literature review section. A critical analysis of the literature review is necessary.
Response: Thank you so much for your comment. A new section (4. Literature review) has been added to cover the historical perspective of the study in the literature and to bring out the literature that was initially ignored.
Comments and Suggestions 3: Specify the theoretical background of the paper, especially the hypothesis.
Response: Thank you so much for your comment. the theoretical background of the paper with the hypothesis is stated in section 5.1.
Comments and Suggestions 4: Results of some sections are presented mechanically. Cross Connection is necessary (Is your finding similar to or contradicts other studies? If so why?)
Response: Thank you so much for your comment. The problem here is that the literature on the relationship between LSLA and farm investment decisions is scanty. Nonetheless, we have tried to support our findings with a few existing studies on the subject.
Comments and Suggestions 5: Please explain why “On the contrary, level of investment in the presence of tenure security is higher for households that are not directly or indirectly exposed to LSLA by domestic and foreign entities. These results, therefore, imply that in the absence of LSLA, an increase in perception of tenure security tends to enhance the level of investment in land-improving techniques. However, the emergence of LSLA will affect such perception and this will, in turn, reduce level of investment in land-improving techniques.”
Response: Thank you so much for your comment. We have now explained the above finding as follows:
“Tenure insecurity brings about uncertainty in the farmer’s mind as to whether he/she may be able to reap full benefits of this/her investment. LSLA remains one of the central issues causing tenure insecurity because of the displacements associated with it. Thus, tenure security and absence of LSLA among households is therefore likely to enhance investment in land improving techniques as observed in the results”.
Comments and Suggestions 6: This manuscript requires proper proofreading and grammar checking.
Response: Thank you so much for this comment. The article has been checked for typos and grammatical errors using Grammarly.
Thank you so much.

Round 2
Reviewer 3 Report
1. It is recommended that literature citations be streamlined and that the most recent, important, and classic literature be cited. Such as "In an attempt to resolve the controversies surrounding LSLA, several empirical studies investigated its implications on livelihoods (e.g., Adams et
al., 2019; Adewumi et al., 2013; Agbley, 2019; Aha & Ayitey, 2017; Ahmed et
al., 2018; Baumgartner et al., 2015; Boamah, 2014; Boamah & Overa, 2015;
Bottazzi et al., 2018; Darkwah & Medie, 2017; Deininger & Xia, 2016;
Giovannetti & Ticci, 2016; Gyapong, 2019, 2020; Herrmann & Grote, 2015;
Herrmann, 2016; Jiao et al., 2015; Kidido & Kuusaana, 2014; Nolte & Ostermeier, 2017; Nyantakyi-Frimpong & Kerr, 2016; Shete & Rutten, 2015;
Williams et al., 2012; Yengoh & Armah, 2015; etc.). " It is recommended that 50-70 citations are sufficient for a paper.
2. Note that the footnotes, tables, and text are in the same font. Such as Table 2, and Table4.
3. It is recommended to streamline the language used, which is now too redundant.
4. Note the deletion of lines 696-697 of “6. Conclusions”.
Author Response
Response to Reviewer 3 Comments and Suggestions for Authors
Comments and Suggestions 1: It is recommended that literature citations be streamlined and that the most recent, important, and classic literature be cited. Such as "In an attempt to resolve the controversies surrounding LSLA, several empirical studies investigated its implications on livelihoods (e.g., Adams et al., 2019; Adewumi et al., 2013; Agbley, 2019; Aha & Ayitey, 2017; Ahmed et al., 2018; Baumgartner et al., 2015; Boamah, 2014; Boamah & Overa, 2015; Bottazzi et al., 2018; Darkwah & Medie, 2017; Deininger & Xia, 2016; Giovannetti & Ticci, 2016; Gyapong, 2019, 2020; Herrmann & Grote, 2015; Herrmann, 2016; Jiao et al., 2015; Kidido & Kuusaana, 2014; Nolte & Ostermeier, 2017; Nyantakyi-Frimpong & Kerr, 2016; Shete & Rutten, 2015; Williams et al., 2012; Yengoh & Armah, 2015; etc.). " It is recommended that 50-70 citations are sufficient for a paper.
Response: Thank you so much for your comment. We are grateful. The references are now reduced to 69 citations. We have also reduced and updated the above citations with the most recent literature. Please see lines 62-65 of the paper for consideration. The outcome is also below for consideration:
“In an attempt to resolve the controversies surrounding LSLA, several empirical studies investigated its implications on livelihoods (e.g., Adams et al., 2019; Atuoye et al., 2021; Borras et al., 2022; Gyapong, 2020; Rasva & Jürgenson, 2022).”
Comments and Suggestions 2 Note that the footnotes, tables, and text are in the same font. Such as Table 2, and Table4.
Response: All the footnotes, tables, and text are now presented in the same font. Thank you.
Comments and Suggestions 3: It is recommended to streamline the language used, which is now too redundant.
Comments and Suggestions 4: Note the deletion of lines 696-697 of “6. Conclusions”.
Response: Please, “6. Conclusions” has been deleted. Thank you so much.
